# Orthosteric STING inhibition elucidates molecular correction of SAVI STING

Tao Xie [1] ✉, Max Ruzanov[2], David Critton[2], Leidy Merselis[3], Joseph Naglich[4], John S. Sack [2], Ping Zhang[5], Chunshan Xie[4], Jeffrey Tredup[5], Laurel B. Stine [3], Cameron Messier[3], David L. Hope[3], Janet Caceres-Cortes[1], Luciano Mueller[1], Alaric J. Dyckman[6], John A. Newitt [5], Asmita Choudhury[7] & Stephen C. Wilson[3] ✉

While the progression of STING activators into the clinic has been successful, the discovery and clinical progression of STING inhibitors remain elusive. Questions persist about the molecular properties needed to distinguish between a STING activator and inhibitor, particularly within SAVI disease, a monogenic autoinflammatory disease that renders STING constitutively active, and how different conformations correlate to function. In this work, we use an orthosteric STING activator and inhibitor from the same chemical series to discover that STING M271 is a critical residue for molecular activation that can be leveraged as a unique molecular signature for pharmacological or genetically driven activation and inhibition. Furthermore, we demonstrate how the therapeutic requirements of a molecular corrector of SAVI STING differs from an orthosteric STING inhibitor, and why this is important for the SAVI disease population.

STING is broadly implicated in various diseases, including cancer, autoimmune disorders, neurodegeneration, and rare monogenic diseases[1]. Early drug discovery campaigns focused on STING activation as a promising platform for cancer immunotherapy yet failed in multiple clinical trials due to lack of efficacy thus far[2]. Current research and development activities concentrate on STING inhibition for treating autoimmune disease and neuroinflammation. While the progression of STING activators into the clinic has been successful, the discovery and clinical progression of STING inhibitors remain elusive. Questions persist about the molecular properties needed to distinguish between a STING activator and inhibitor, particularly within SAVI disease, a monogenic autoinflammatory disease that renders STING constitutively active[3], and how different conformations correlate to function.

In this work, we use an orthosteric STING activator and inhibitor from the same chemical series to discover that STING M271 is a critical residue for molecular activation. The M271[CH3] NMR chemical shifts reveal a unique molecular signature for pharmacological or genetically driven activation and inhibition that is not able to be resolved by other biophysical methods. We show hydrophobic interaction in this region plays an important role in regulating STING activity. The findings also explain how gain-of-function mutations V155M and G158A activate STING in a ligand-independent way by shifting the conformation of STING from the inactive state towards the active state. Additionally, M271[CH3] is proximal to the most common SAVI mutation, V155M, and using an orthosteric STING inhibitor, we show partial rescue and molecular correction of

[1]Drug Discovery Analytical, Lead Discovery and Optimization, Discovery & Development Sciences, Bristol Myers Squibb, Lawrenceville, NJ 08648, USA. [2]Structural Biology, Lead Discovery and Optimization, Discovery & Development Sciences, Bristol Myers Squibb, Lawrenceville, NJ 08648, USA. [3]Discovery Immunology, Bristol Myers Squibb, 250 Water St, Cambridge, MA 02141, USA. [4]Mechanistic Pharmacology, Discovery & Development Sciences, Bristol Myers Squibb, Lawrenceville, NJ 08648, USA. [5]Protein Science, Lead Discovery and Optimization, Discovery & Development Sciences, Bristol Myers Squibb, Lawrenceville, NJ 08648, USA. [6]Immunology Chemistry, Discovery & Development Sciences, Bristol Myers Squibb, Lawrenceville, NJ 08648, USA. [7]Biocon-Bristol Myers Squibb Research and Development Center, Biocon Park, Plot No. 2 & 3, Bommasandra Phase IV, Jigani Link Road, Bangalore 560099, India. ✉e-mail: tao.xie@bms.com; scwilson@gmail.com

STING V155M. Finally, these data present insights into therapeutic STING molecular correction for treating SAVI patients. Our results elucidate an unappreciated structural interaction critical for STING modulation that could be utilized as a molecular diagnostic tool for drug discovery. Furthermore, we demonstrate how the therapeutic requirements of a molecular corrector differ from an orthosteric STING inhibitor, and why this is important for the SAVI disease population.

## Results

### diABZI agonist and inhibitor bound to STING are indistinguishable

The diABZI chemical family comprises molecules that can activate or inhibit STING signaling[4,5]. To elucidate the differences in molecular architecture between orthosteric STING inhibition and activation, we determined the crystal structures of human STING[155-341] bound to the orthosteric inhibitor, diABZI-i, and the orthosteric activator, diABZI-a1, derived from the diABZI series (Fig. 1a). Both compounds have similar potency for inhibiting and eliciting IFNβ in human PBMCs, respectively ($IC_{50}^{diABZI-i} = 49 \pm 8$ nM, 1 donor | $EC_{50}^{diABZI-a1} = 117 \pm 72$ nM, 3 donors). However, the differences between the two structures are unexpectedly modest with a r.m.s.d. of 0.548 Å$^2$ for all atoms (Fig. 1b, and Supplementary Fig. 1, Supplementary Table 1). Moreover, both compounds adopt similar positioning within the binding pocket with no obvious mechanistic rationale for their reciprocal functions evident in the crystal structures (Fig. 1c). Despite these findings, we analyzed both crystal structures for two characteristic structural features associated with STING activation: beta sheet lid formation and the apical wing distance between STING monomers (Fig. 1d)[6,7].

Beta sheet lid formation over the binding pocket is a well-described signature of cGAMP binding to STING[6]. However, questions have been raised over whether its formation is necessary for driving STING signaling[5]. Comparisons between STING bound to diABZI-i and diABZI-a1 do not reveal any significant differences in lid formation to explain inhibition or activation. Both compounds elicit a disordered STING lid organization that does not extend over the binding pocket (Fig. 1e), which has been previously described in both apo STING and STING bound to cyclic-di-GMP (cdG), a bacteria-derived STING agonist[5,7–10]. Since orthosteric STING inhibition is poorly understood, we compared the STING bound between diABZI-i and THIQi, the only other orthosteric, chemically distinct STING inhibitor reported to date (Fig. 1f)[11]. Our findings show no significant differences in lid formation between these two structures (Fig. 1g, and Supplementary Table 1), substantiating that beta sheet lid organization over the binding pocket is not required for orthosteric STING activation or inhibition.

Another well-reported feature of cGAMP-STING binding is the shortening of the apical wing distance between STING monomers[6,7]. Both diABZI bound structures adopt a splayed open conformation with an apical wing distance of ~47 Å and ~52 Å for diABZI-a1 and diABZI-i, respectively (Fig. 1h, and Supplementary Table 1). Since the ~5 Å difference between the two structures could be significant enough to predict STING inhibition or activation, we assessed these distances relative to reference structures of apo STING[8], cGAMP-STING[6], and cdG-STING[8,9] in the PDB, and the THIQi-STING bound structure (Fig. 1h). All apical wing distances calculated in this study are bookended by cGAMP-STING (~39 Å) and cdG-STING (~56 Å), indicating that apical wing shortening or lengthening is not predictive for STING activation. Both orthosteric STING inhibitors used in this study adopt similar splayed open conformations, suggesting an open conformation could be required for inhibition. Therefore, the totality of our analysis indicates that STING[155-341] crystallography cannot readily distinguish between diABZI-driven STING inhibition and activation.

### NMR of M271[CH3] distinguishes between STING activation and inhibition

Since small molecule binding to STING is expected to drive conformational changes that lead to either signal transduction (agonist) or prevent signal transduction (inhibitor), we hypothesized that structural differences between diABZI-a1 and diABZI-i bound STING[155-341] could be elucidated in the solution-state[12]. To that end, we investigated binding of diABZI-a1 and diABZI-i to STING[155-341] using solution NMR spectroscopy. THIQi (inhibitor) and ABZI (agonist)-bound STING[155-341] led to the highest quality spectra and therefore were used for $^1$H-$^{15}$N HSQC and $^1$H-$^{13}$C HSQC resonance assignments (see "Methods" section for detailed description of backbone and sidechain assignments, Supplementary Figs. 2, 3). Approximately 88% of the backbone amides and 85% of the methyls were assigned for THIQi-bound STING[155-341], while 84% of the backbone amides and 60% of the methyls were assigned for ABZI-bound STING[155-341].

Comparison of $^1$H-$^{13}$C HSQC spectra between diABZI-i and diABZI-a1 revealed multiple differences (Fig. 2a, b). Most notably, the methyl group on the M271 side chain (M271[CH3]) exhibited a significant downfield shift in both proton (0.920 ppm) and carbon (1.217 ppm) dimensions with diABZI-a1 but not diABZI-i. Re-examination of the diABZI-i and diABZI-a1 bound crystal structures exhibited no clear distinction between the two (Fig. 2c) with a r.m.s.d for the M271 residue <1 Å$^2$ (Supplementary Fig. 1a).

Major upfield shifts in both proton (0.721 ppm) and carbon (1.667 ppm) dimensions were also observed for A277[CH3] which is in close proximity to M271 (Fig. 2c) upon binding to diABZI-a1 but not for diABZI-i (Fig. 2b). These shifts are opposite in directionality to the M271[CH3] shifts. Both M271[CH3] and A277[CH3] are >10 Å from the binding pocket as measured in the respective crystal structures. However, the magnitude and resolution of the M271[CH3] shift compared to A277[CH3] as well as its through-space proximity (2.5 Å) to V155 (Fig. 2c) caused us to focus our attention on M271[CH3].

Since the differences in M271[CH3] shifts between diABZI-i and diAZI-a1 were significant in magnitude and unobstructed by other correlations, we investigated whether this was a generalizable effect among other orthosteric STING inhibitors and agonists. We compared the STING-bound $^1$H-$^{13}$C HSQC spectra among an additional seven chemically diverse STING agonists including cGAMP[13–16], cdG[10], ABZI[5], diABZI-a2[5], MSA-2[17], and BMS-025[18,19] as well as THIQi (Fig. 2e). All STING agonists displayed significant downfield proton and carbon shifts (Fig. 2e, f, and Supplementary Fig. 4a, b). On average, STING agonists elicited a downfield proton shift of 1.016 ppm (min = 0.92 ppm, max = 1.131 ppm) and an average downfield carbon shift of 1.462 ppm (min = 1.18 ppm, max = 1.751 ppm). The observation of two resonances of M271[CH3] upon binding to BMS-025 and cGAMP is caused by binding asymmetry. On the other hand, both STING inhibitors showed no appreciable effect on the proton chemical shift on average (-0.004 ppm) and a small average upfield carbon shift of -0.122 ppm (Fig. 2e, f, and Supplementary Fig. 4a, b). Similarly, A277[CH3] underwent large proton and carbon shifts but in upfield direction upon binding to agonists compared to binding to inhibitor (Supplementary Fig. 4b, c).

Since M271[CH3] and A277[CH3] are >10 Å from the binding pocket, it is likely that the large chemical shift movements are caused by agonist binding-induced allosteric conformational changes rather than direct interaction. By mapping the chemical shift differences between STING-bound ABZI and THIQi onto crystal structures, we found that the hydrophobic region harboring V155, M271 and A277 beneath the CDN binding site is significantly perturbed (Fig. 3a–d), indicative of distinct conformations between agonist and antagonist-bound STING[155-341].

## Conformation in dimer interface determines STING functional state

To understand why agonist binding induced such large shifts but not inhibitors, we compared crystal structures of apo-STING (PDB: 4EMU) and THIQi-bound STING with 2′,3′-cGAMP-bound STING. In the inactive state (i.e. apo and THIQi-bound STING), M271$^S$ forms sulfur-aromatic interactions with W161 and F279. M271$^{CH3}$ sits directly in between the aromatic rings of W161 and F279, which leads to a large shielding effect caused by the aromatic ring current and consequent upfield shift. (Fig. 3e, f). A277$^{CH3}$ resides above, off center the indole ring of W161 and proximal and parallel to the benzene ring of Y274 (Fig. 3e, f). In the active state (i.e. cGAMP-bound STING), the sulfur-aromatic interaction is disrupted due to a

shift in relative orientation between M271$^{CH3}$ and the aromatic rings of W161 and F279 (Fig. 3g). M271$^{CH3}$ experiences deshielding by moving outside the indole rings of W161 yet remains in the same plane as the benzene ring of F279 (Fig. 3g), leading to its downfield shift. In comparison, A277$^{CH3}$ experiences large shielding by moving to the top center of the W161 indole ring and remains distal from the Y274 ring, resulting in upfield shifts. These local conformational differences between active and inactive state clearly explain the respective large downfield and upfield shifts of M271$^{CH3}$ and A277$^{CH3}$ upon binding to agonists and the small chemical shift changes upon inhibitor binding. Therefore, our NMR data shows that other than forming the beta sheet lid structure and adopting the closed conformation, STING LBD (ligand binding domain) experiences a

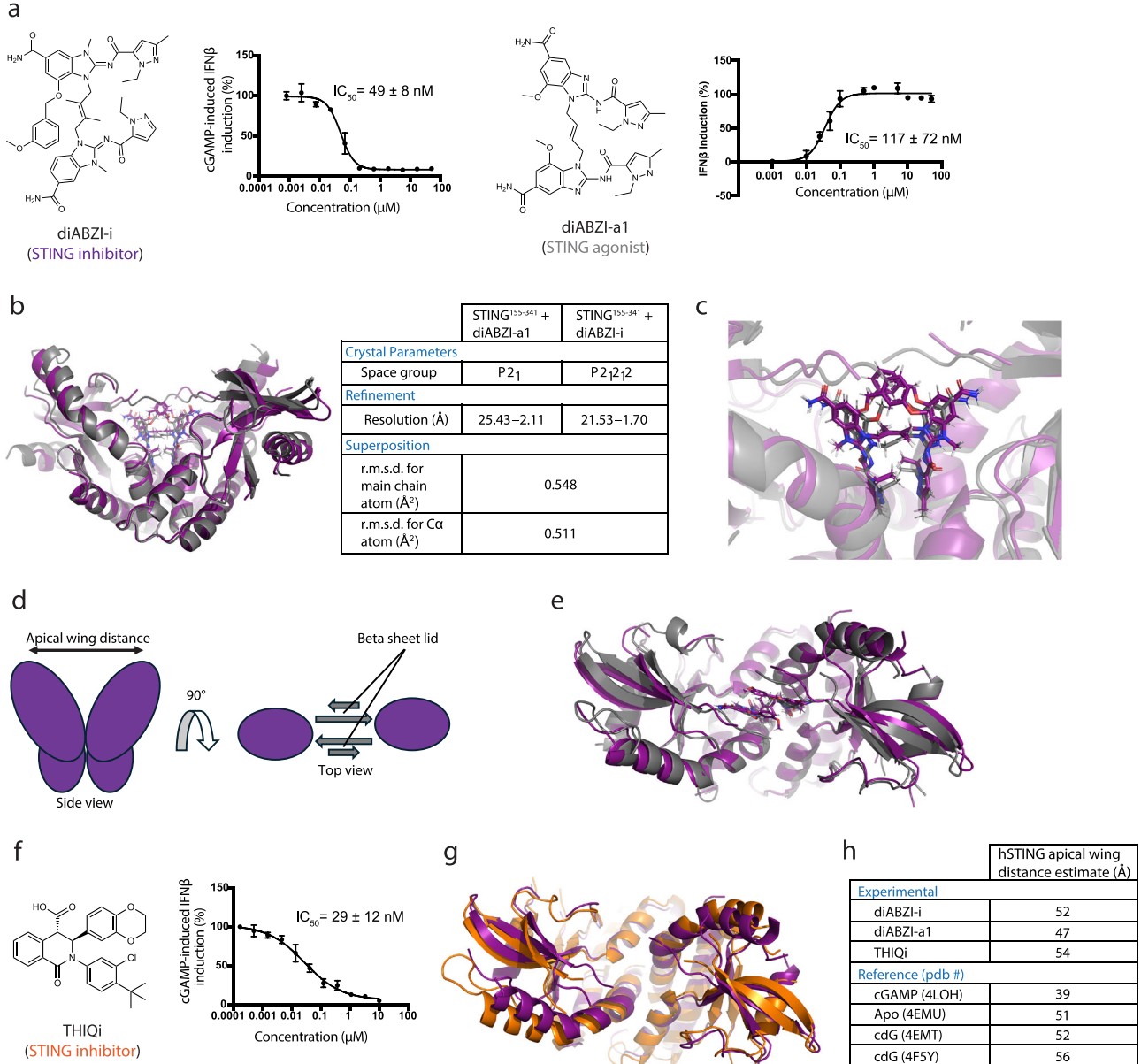

**Fig. 1 | STING bound diABZI-based agonist and inhibitor are nearly indistinguishable in crystal structures. a** diABZI-i inhibits cGAMP-induced IFNβ in PBMCs (n = 3 | 1 donor) and diABZI-a1 induces IFNβ in PBMCs (n = 3 | 3 donors, representative donor displayed). Data represented as mean with SD in graph. Source data are provided as a Source Data file. **b** Superposition of crystal structures (side view) of STING bound to diABZI-a1 (gray) and diABZI-i (purple). **c** diABZI-a1 and diABZI-i superposition over the STING binding pocket. **d** Cartoon depicting

apical wing distance and beta sheet lid between STING monomers. **e** Superposition of crystal structures (top view) of STING bound to diABZI-a1 (gray) and diABZI-i (purple). **f** THIQi inhibits cGAMP-induced IFNβ in PBMCs (n = 3 | 1 donor). Data represented as mean with SD in graph. Source data are provided as a Source Data file. **g** Crystal structure overlay (top view) of STING bound to THIQi (orange) and diABZI-i (purple). **h** Apical wing distance measurements from structures in this study and reference structures collected from the PDB.

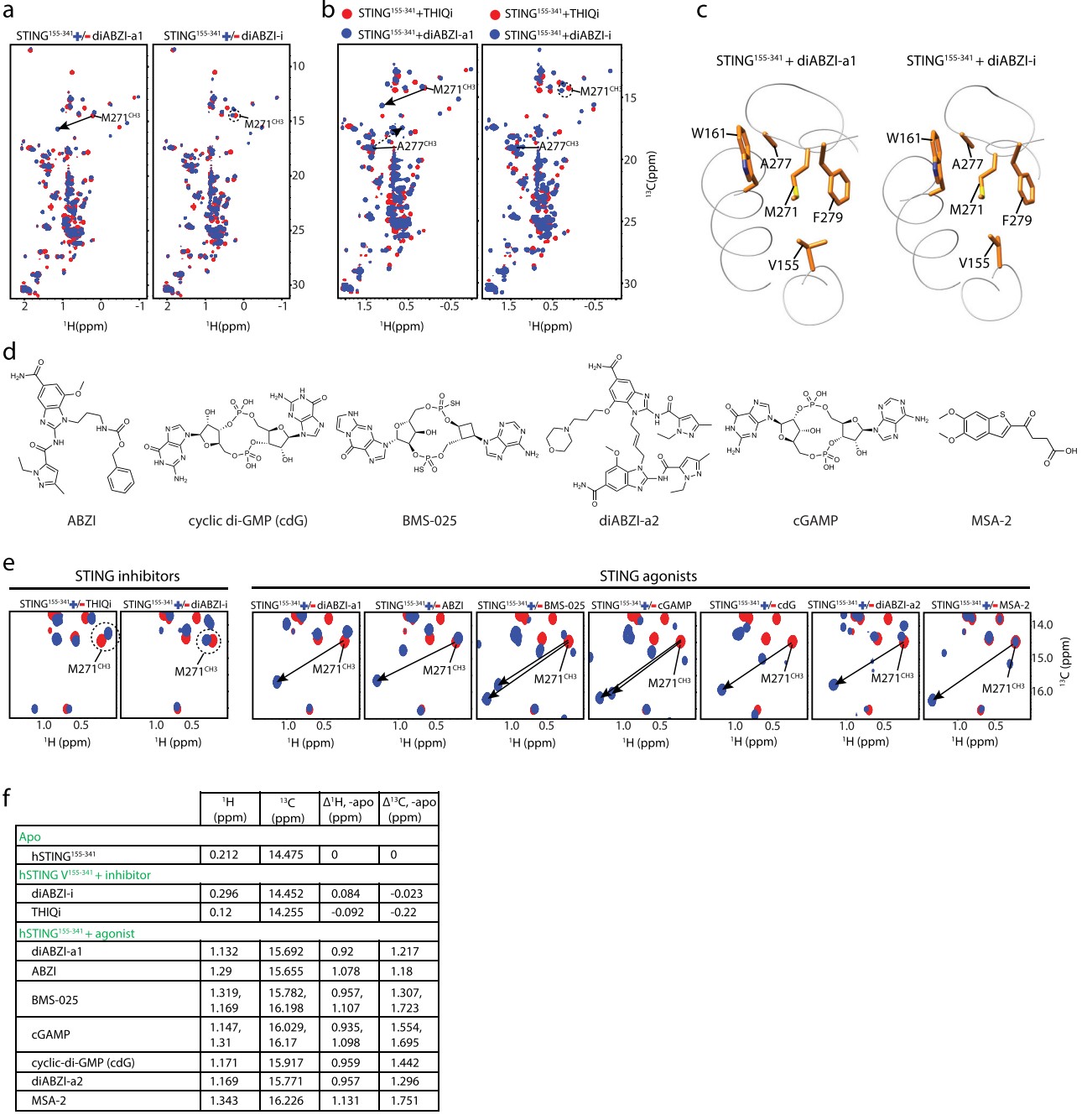

**Fig. 2 | NMR distinguishes between STING activation and inhibition via M271$^{CH3}$ chemical shift in $^1$H-$^{13}$C HSQC spectrum. a** Overlay of $^1$H-$^{13}$C HSQC spectra of STING$^{155\text{-}341}$ with and without diABZI-a1 and diABZI-i. **b** Overlay of THIQi-bound STING$^{155\text{-}341}$ with diABZI-a1 and diABZI-i-complexed STING$^{155\text{-}341}$. **c** Crystal structures focused on V155 and M271 residues of STING$^{155\text{-}341}$ bound with diABZI-a1 and diABZI-i. **d** Chemical structures of diverse STING agonists used for this study. **e** Overlay of $^1$H-$^{13}$C HSQC of STING$^{155\text{-}341}$ recorded in the absence (red) and presence (blue) of various antagonists and agonists highlighting M271$^{CH3}$. **f** Tabulated M271$^{CH3}$ $^1$H and $^{13}$C chemical shift differences between apo and bound STING.

conformational change in the hydrophobic region of the dimer interface upon 2′,3′-cGAMP binding. Resonances of M271$^{CH3}$ and A277$^{CH3}$ are at a similar position in the $^1$H-$^{13}$C-HSQC spectra for all agonist-bound STING$^{155\text{-}341}$ (Fig. 2e and Supplementary Fig. 4), suggesting the conformation observed in our 2′,3′-cGAMP-bound STING$^{155\text{-}241}$ crystal structure (Fig. 3g) exists in other agonist-bound STING and represents the real active conformation in solution. Therefore, agonist binding allosterically induces conformational changes in the hydrophobic region, which includes V155, M271 and A277.

## SAVI mutations shift conformational equilibrium to the active state

The STING V155M mutation is reported to be the most common genotype driving STING-Associated Vasculopathy with onset in Infancy (SAVI), a severe autoimmune disease with poor treatment options available[3,20]. Approximately 62% of known SAVI patients harbor the V155M mutation, and ~86% of reported SAVI patients have mutations in the same region as V155M, highlighting a crucial area for eliciting STING activation[20]. The through-space proximity of M271 to V155 (2.5 Å) suggests an important relationship between these two residues

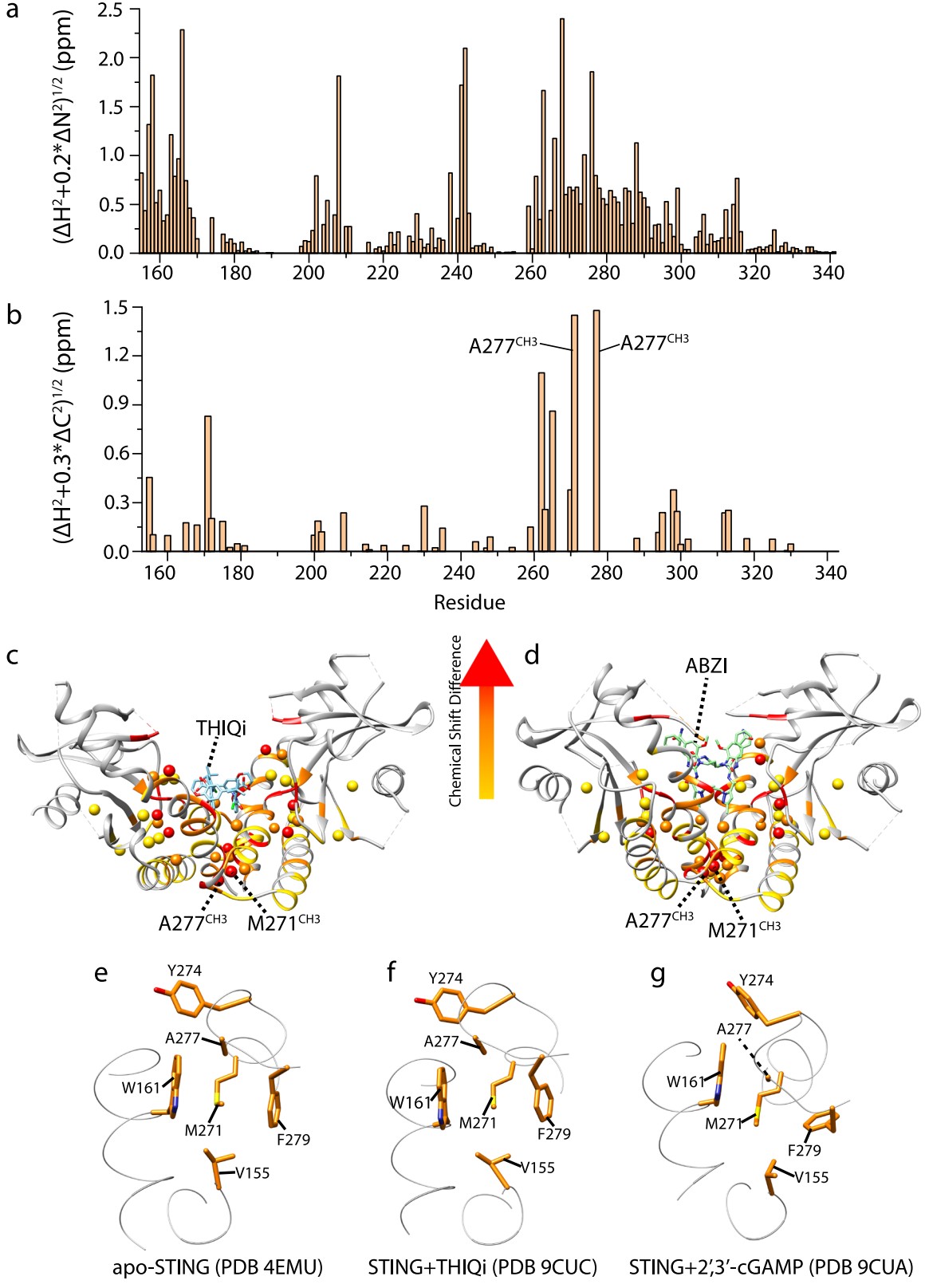

**Fig. 3 | Structural basis for large chemical shift changes of M271$^{CH_3}$ and A277$^{CH_3}$ upon agonist binding.** Backbone amide (**a**) and methyl (**b**) chemical shift differences between STING$^{155-341}$-THIQi and STING$^{155-341}$-ABZI. Chemical shift differences are mapped onto (**c**) STING$^{155-341}$-THIQi and (**d**) STING$^{155-341}$-ABZI structures. M271$^{CH_3}$ and A277$^{CH_3}$ exist underneath the binding pocket and are distant from the CDN binding pocket yet show significant chemical shifts differences, indicating different conformations between these two complexes. Hydrophobic interaction network in (**e**) apo-STING$^{155-341}$, (**f**) THIQi-bound STING$^{155-341}$, and (**g**) 2', 3'-cGAMP bound STING$^{155-341}$.

(Fig. 2c), and the phylogenetic conservation of M271 between invertebrates and vertebrates indicates a preservation of its importance over the evolution of STING signaling (Fig. 4a). Structural analysis also suggests an important interaction between V155M and M271 that reinforces STING dimer stability[21]. Therefore, we sought to investigate whether the M271[CH3] NMR signature for pharmacologically induced STING activation is also observed in genetically induced STING activation through SAVI mutations.

We prepared $^{13}$C, $^{15}$N labelled STING$^{155-341}$ V155M and G158A[22,23] for solution-state NMR spectroscopy. Both SAVI mutations elicited significant M271 downfield chemical shift changes consistent with those illustrated with pharmacological activation (Fig. 4b). STING$^{155-341}$ V155M elicited a downfield proton shift of 0.701 ppm and a downfield carbon shift of 1.69 ppm, whereas STING$^{155-341}$ G158A caused a downfield proton shift of 1.328 ppm and a downfield carbon shift of 2.233 ppm (Fig. 4c). We also examined the NMR profile of STING$^{155-341}$ G158E, a rationally designed mutation at a SAVI location that does not elicit constitutive STING signaling[23]. However, the M271[CH3] peak in the $^1$H-$^{13}$C HSQC spectrum could not be identified (Supplementary Fig. 5), which may be due to STING destabilization from steric bulk or negative charge as previously suggested[23]. These data shows that the M271[CH3] chemical shift can be leveraged as a diagnostic signature for pharmacological or genetic STING activation.

To further elucidate that the V155M mutation functions by perturbing conformational equilibrium between the inactive and active states, we performed an NMR competition experiment. We compared NMR titration of STING$^{155-341}$-THIQi complex to titration of STING$^{155-341}$ V155M-THIQi complex with agonist BMS-025. M271[CH3] chemical shifts of THIQi-bound STING$^{155-341}$ and THIQi-bound STING V155M$^{155-341}$ remain in the upfield region in both spectra (Fig. 4d, f), indicating both STING$^{155-341}$ and STING$^{155-341}$ V155M adopt the inactive conformation in the presence of inhibitor. Addition of the same amount of agonist BMS-025 led to appearance of new peaks of M271[CH3] at the downfield position which represent active conformation in the spectra of both samples (Fig. 4e, g) as a result of competition between inhibitor and agonist for the CDN binding pocket. Importantly, the relative intensity of the downfield new peak to the original peak in spectrum of V155M mutation is much larger than that in the wild-type, suggesting inhibitor binds to the mutant with a lower affinity compared to binding to the wild-type, and agonist binds to the mutant with a higher affinity than binding to the wild-type due to the energy barrier between the active and the inactive states. Therefore, inhibiting SAVI STING by targeting the CDN binding pocket represents a bigger challenge than suppressing the wild-type STING. This data further indicates that SAVI gain-of-function mutations V155M and G158A activate STING by stabilizing a conformation similar to that adopted by agonist-bound STING in the dimer interface, and M271[CH3] chemical shift can be leveraged as a diagnostic signature for regulation states of STING.

## Effect of M271 mutagenesis on interferon signaling

The relationship between M271 and SAVI V155M is a provocative connection that we wanted to leverage to gain insight into SAVI for its potential treatment. As such, we decided to evaluate the consequence of M271 mutagenesis on constitutive STING interferon (IFN) signaling with and without a V155M background. Except for M271I, M271 mutagenesis to alternative hydrophobic amino acids, M271A, M271I, M271L, and M271V, conferred statistically significant constitutive activation to STING on par with V155M-driven signaling (Fig. 4h, and Supplementary Fig. 6). Our functional assay indicates that M271 critically controls signaling induced by the V155M mutation. M271 mutagenesis to M271A, M271I and M271V controlled the amplitude of V155M induced signaling. M271L is distinctive in that its activity under a V155M background is enhanced compared to M271L alone, yet M271L + V155M is attenuated compared to V155M alone. Furthermore, M271S and M271G completely ablated V155M-driven signaling, confirming that M271 is a

critical residue for V155M SAVI signal transduction (Fig. 4h, and Supplementary Fig. 6).

M271 mutagenesis can confer a gain-of-function state suggesting that mutation of this residue could confer SAVI-like disease. While no clinical publication has been described to date on such a patient population, we identified one heterozygous carrier of M271I within the gnomAD database[24]. It is currently unknown whether this carrier presents with SAVI-like phenotypes, but it is notable to mention that Polyphen[25] and SIFT[26] annotate M271I as a "possibly damaging" and "deleterious" mutation, respectively.

## Molecular correction of V155M SAVI

Given that the M271[CH3] chemical shifts act as a diagnostic for STING activation, we decided to examine whether orthosteric STING inhibition could molecularly correct the M271[CH3] activation shift of V155M SAVI[27,28]. Both diABZI-i and THIQi showed partial upfield corrections of the STING V155M M271[CH3] chemical shift (Fig. 5a, b). diABZI-i exhibited a $\Delta^1$H = -0.284 ppm, $\Delta^{13}$C = -0.748 ppm correction, and THIQi led to a $\Delta^1$H = -0.47 ppm, $\Delta^{13}$C = -0.514 ppm correction (Fig. 5c). All STING agonists led to further downfield proton and carbon shifts of V155M M271[CH3] (Fig. 5c) except for ABZI, where only a downfield proton shift was apparent with no significant change in the carbon chemical shift.

SAVI patient carriers present with symptoms consistent with a type I interferonopathy. The disease often manifests in early life leading to systemic inflammation and a high mortality rate with features including elevated type I IFN, skin vasculopathy, arthritis, and pulmonary disease[3,20]. The pulmonary aspect of SAVI disease is the most common feature among patients and a key driver for mortality. In published studies to date, JAK inhibitors such as ruxolitinib are the most reported treatment for SAVI[3,29–31]. However, it is important to note that while treatment with ruxolitinib or other JAK inhibitors is reported to improve clinical disease scores, reports indicate little to no impact on clinical IFN scores[20,29,31]. While more work is needed to understand this paradox, it implies that targeting STING could be the only critical node for therapeutically resolving SAVI disease.

Since THIQi and diABZI-i show partial molecular correction of SAVI V155M STING by M271 in the $^1$H-$^{13}$C HSQC spectrum, we were interested in understanding whether orthosteric STING inhibition could also reverse V155M SAVI constitutive activation in a cellular model. A stably expressed STING V155M THP-1 model exhibits elevated IFN ( >10x) over WT THP-1 in the absence of cGAMP stimulation (Fig. 5d). THIQi demonstrates partial rescue of the elevated IFN in this model (IC$_{50}$ = 13 ± 4 μM, % inhibition = 54% ± 26%) with an inhibition potency within three-fold of the WT THP-1 cGAMP stimulation model (Fig. 5e, f). In contrast, diABZI-i demonstrates expected inhibition in the WT THP-1 cGAMP stimulation model (IC$_{50}$ = 3 ± 2 μM) but potent agonism in the V155M SAVI model (EC$_{50}$ = 17 ± 7 nM) with a > 170x increase in potency relative to inhibition potency in WT THP-1s. These data suggest orthosteric STING inhibition may not necessarily resolve the constitutive signaling associated with V155M SAVI, and in fact cautions that an orthosteric STING inhibitor in WT carriers could serve as a STING agonist in SAVI carriers, thereby exacerbating disease.

## Discussion

Cryo-EM studies of full-length STING have examined the important interplay between the cyclic dinucleotide (CDN) binding region and transmembrane region of STING, leading to oligomerization-dependent effects on signaling[22,23,32–34]. However, STING has multiple signaling mechanisms embedded and distributed throughout its architecture, which makes therapeutic inhibition challenging[35–38].

Since all agonists induced downfield shifts of M271[CH3], it is intriguing whether the conformation captured in our crystal structure of STING$^{155-34}$ -2',3'-cGAMP complex (Fig. 3g) can be seen in other

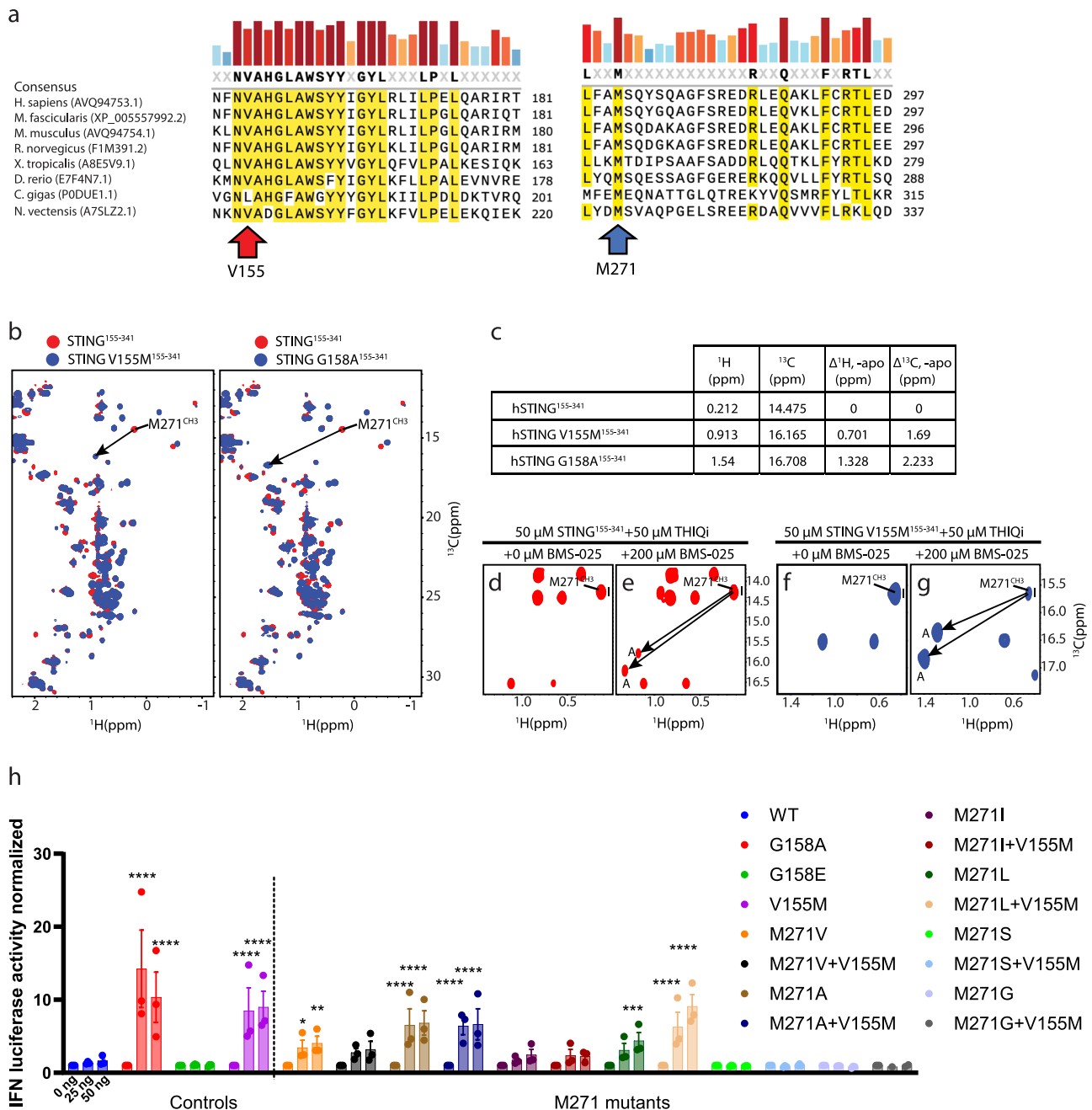

**Fig. 4 | The evolutionary conserved M271 residue mediates SAVI STING V155M-driven IFN induction. a** V155 and M271-focused phylogenetic sequence alignment. **b, c** Comparison of $^1$H-$^{13}$C HSQC spectra among STING WT, SAVI STING V155M and SAVI STING G158A. $^1$H-$^{13}$C HSQC of STING$^{155-341}$-THIQi in the absence (**d**) and presence (**e**) of BMS-025 and of STING V155M$^{155-341}$-THIQi in the absence (**f**) and presence (**g**) of BMS-025. I and A in d-g represent the inactive and active conformations, respectively. **h** M271-focused mutational analysis using HEK293 IFN-driven luciferase reporter cell line with variable amounts of plasmid transiently transfected ($n = 3$ biological replicates). Data represented as mean with SEM in graph. Source data are provided as a Source Data file. Statistical significance determined by two-way ANOVA against WT with adjusted $p$ value depicted (*- $p \leq$ 0.05, **- $p \leq$ 0.01, ***- $p \leq$ 0.001, ****- $p \leq$ 0.0001). M271V (25 ng), $p = 0.0357$. M271V (50 ng), $p = 0.0061$. M271L (50 ng), $p = 0.0007$.

agonist-bound STING structures. We examined conformations in the dimer interface of diABZI-i-bound STING and STING in complex with different agonists (Fig. 6 and Supplementary Table 2). We designate the conformation similar to that observed in our THIQi-complexed STING$^{155-341}$ structure and STING LDB (PDB 4EMU) as state 1 (S$_1$) (Fig. 6a) and the conformation seen in our 2′,3′-cGAMP-bound STING$^{155-341}$ structure as state 2 (S$_2$) (Fig. 6b). Based on our NMR results and structural analysis, S$_1$ and S$_2$ states correlate well with the upfield position of M271$^{CH3}$ observed in apo STING$^{155-341}$ as well as inhibitor-bound STING$^{155-341}$ (Fig. 2e) and downfield position of

M271$^{CH3}$ seen in agonist-complexed STING$^{155-341}$ (Fig. 2e), SAVI mutations V155M, and G158A (Fig. 4b), respectively in $^1$H-$^{13}$C HSQC spectra. Surprisingly, either S$_1$ or S$_2$ conformations can be captured in the crystal structures of agonist-bound STING LDB (Fig. 6a, b), indicating the conformations of the dimer interface captured in the crystal structure are not always in line with STING functional state. Instead, the NMR chemical shift of M271$^{CH3}$ can serve as a probe to characterize the regulation state of STING.

Although cdG acts as an agonist of STING, its binding to STING does not induce the formation of a beta sheet lid and a

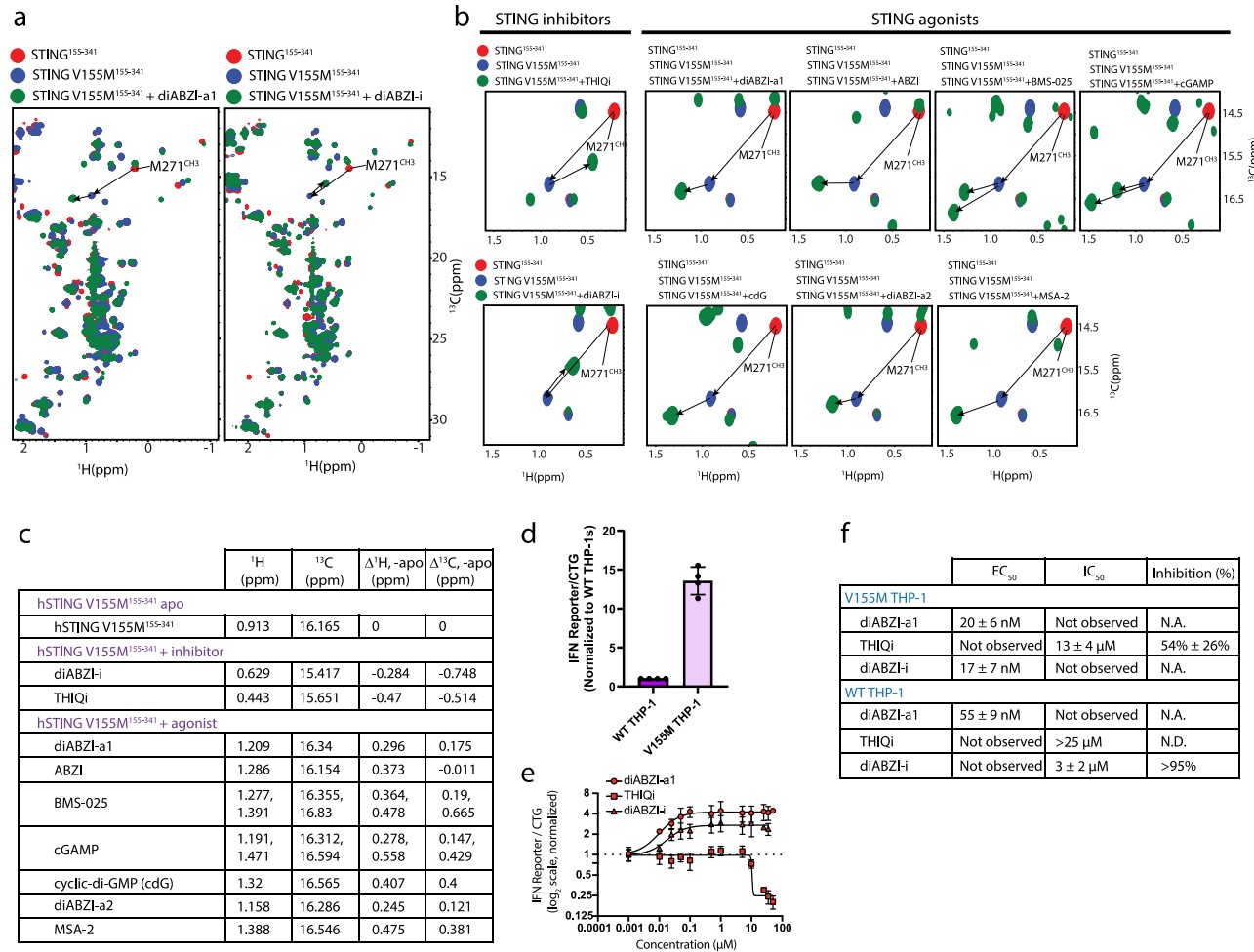

**Fig. 5 | M271^CH3 chemical shift correction is necessary but not sufficient for STING molecular correction of V155M SAVI. a**, **b** Overlay of ¹H-¹³C HSQC of STING¹⁵⁵⁻³⁴¹ (red) and STING¹⁵⁵⁻³⁴¹ V155M recorded in the absence (blue) and presence (green) of various antagonists and agonists. **c** Tabulated M271^CH3 ¹H and ¹³C chemical shift differences between apo and bound STING V155M. **d** Comparison of basal IFN-driven luciferase levels between WT THP-1 and STING V155M THP-1 cell lines (*n* = 4 biological replicates). THP1-Dual™ KI STING (Invivogen, cat# thpd-m155) harbor either the WT allele or V155M point mutation and have the endogenous STING alleles knocked out. Data represented as mean with SD in graph. Source data are provided as a Source Data file. **e** Agonism and antagonism behavior of diABZI-a1, THIQi, and diABZI-i on basal IFN levels in STING V155M THP-1 cell line (*n* = 3 biological replicates); representative replicate shown. Data represented as mean with SD in graph. Source data are provided as a Source Data file. **f** Tabulated IFN-driven luciferase EC₅₀ and IC₅₀ values in WT THP-1 (with and without cGAMP stimulation) and STING V155M THP-1 (no stimulation).

closed conformation in four out of five cdG-bound STING LDB crystal structures (PDB 4EF4, 4F9G, 4EMT and 4F5Y), in which the x-ray crystallography captures a conformation similar to S₁ in the dimer interface (Fig. 6a). However, in the other cdG-bound STING LDB structure (PDB 4F5D), cdG binding leads to a closed conformation and the formation of a beta sheet lid, and the dimer interface remains in a conformation similar to S₂ (Fig. 6b), which is consistent with that predicted from the M271^CH3 chemical shifts. Similarly, both S₁ and S₂ conformations are observed in the crystal structures of diABZI-complex STING LDB (Fig. 6a, b).

However, M271^CH3 can also remain in a slightly different chemical environment from those observed in S₁ and S₂ conformations in the crystal structures of STING LDB bound to both agonists and inhibitor diABZI-i (Fig. 6c). It differs from the S₁ conformation only in the relative position of M271^CH3 to W161 which is similar to that seen in S₂. Instead of sitting on top of W161 indole ring, M271^CH3 moves away from the center top of the indole ring. We denote it as the state 3 (S₃), which may represent an intermediate conformation between S₁ and S₂. S₃ is observed in crystal structures of STING LDB bound to inhibitor diABZI-i and agonists 2',3'-cGAMP, CDA, diABZI, MSA-2, and MK-1454 (Fig. 6c).

Interestingly, all structures with the dimer interface captured in S₁ retain the open conformation without the formation of the beta sheet lid (Fig. 6a and Supplementary Table 2). In comparison, STING structures with the dimer interface adopting S₂ remain in the closed conformation (Fig. 6b and Supplementary Table 2). However, STING structures with the dimer interface in S₃ adopt either open or closed conformation (Fig. 6c and Supplementary Table 2). The correlations between the dimer interface conformation and the open/closed conformation suggests that the dimer interface conformation can be predicted from the chemical shift of M271^CH3 and is crucial for STING regulation.

Cryo-EM structure of chSTING without the c-terminal tail (CCT) reveals STING stays in an auto-inhibited state by forming a bilayer assembly packed in a head-to-head and side-by-side way[32]. The side-by-side packing is mainly mediated by LBDα2-loop-LBDα3 and is crucial for keeping STING in an autoinhibited state. The hydrophobic interactions in the dimer interface were suggested to play an important role in maintaining the configuration of LBDα2-loop-LBDα3[32]. Therefore, the conformational change in this hydrophobic region induced either by agonist binding or mutation could regulate STING function by perturbing the

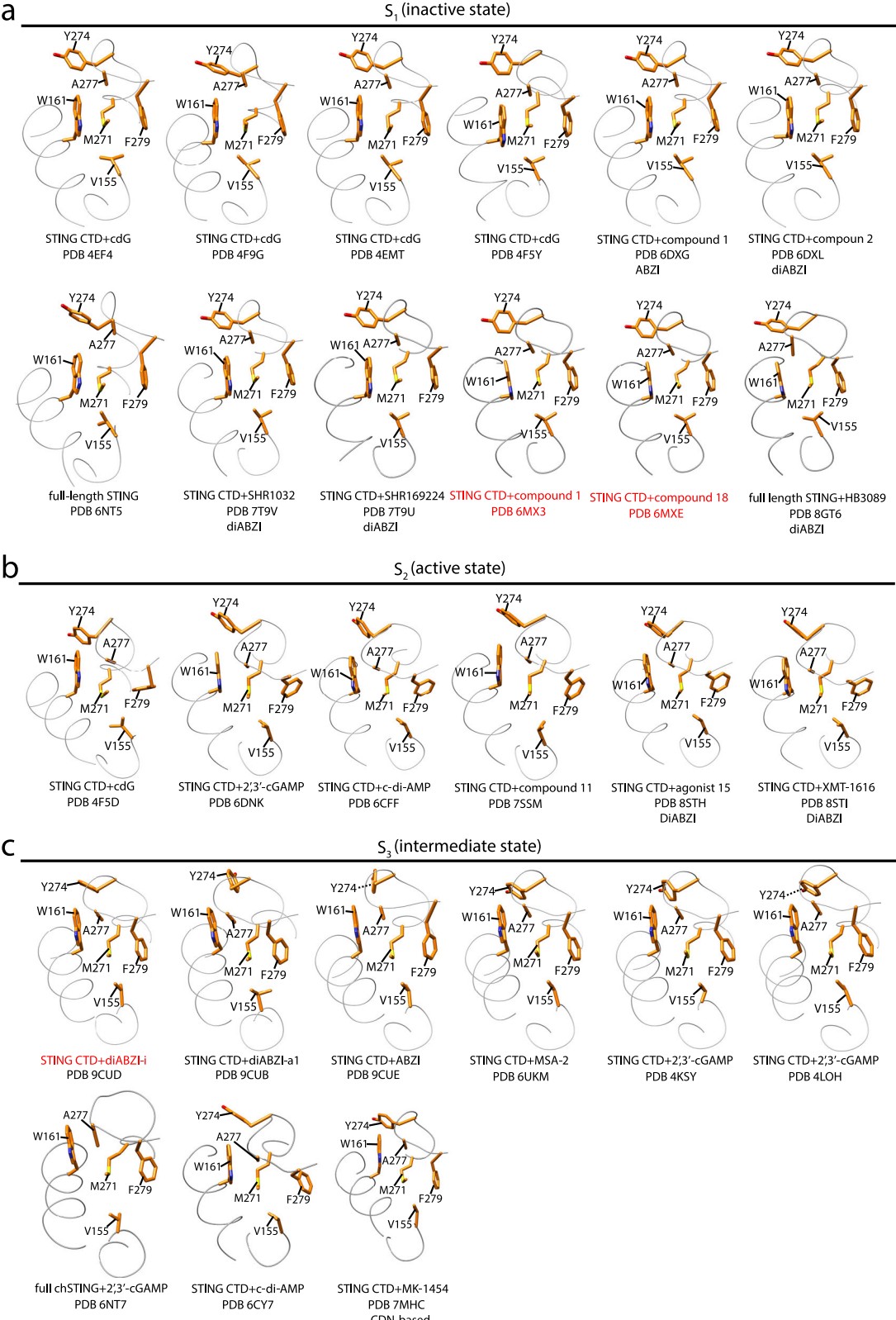

**Fig. 6 | Conformational states of the hydrophobic dimer interface captured in different STING structures. a** State 1 (S$_1$) adopts a similar conformation to that seen in THIQi-complexed STING$^{155\text{-}341}$ (Fig. 3f) and STING LDB (PDB 4EMU, Fig. 3e). **b** State 2 (S$_2$) is similar to that observed in 2′, 3′-cGAMP-bound STING$^{155\text{-}341}$ structure (Fig. 3g). **c** State 3 (S$_3$) remains a conformation between S$_1$ and S$_2$. Compounds highlighted with red color are inhibitors, and others are agonists.

structure of LBDα2-loop-LBDα3. Future studies could explore the correlation between the magnitude of mutation-induced chemical shift of M271$^{CH3}$ and binding affinity of STING to downstream proteins such as TBK1.

The chemical shift of M271$^{CH3}$ elucidates STING signaling and provides a robust diagnostic for STING activation, inhibition, and molecular correction conferred by both genetics and pharmacology. In the case of V155M SAVI, this has exposed the challenges

and potential risks with treating V155M SAVI patients using orthosteric STING inhibitors. In fact, orthosteric STING targeting of V155M SAVI requires molecular correction not just inhibition.

## Methods

### Ethics statement

All research contained with this article complies with rules and regulations maintained by Bristol Myers Squibb. All procedures involving human samples complied with ethical standards and guidelines approved by the Western Copernicus Group Institutional Review Board (WCGIRB) under "Standard Operating Procedure for Obtaining Venous Blood and Other Non-Invasive Biological Specimens for Research Purposes for Bristol-Myers Squibb Research" (WIRB protocol #20161208). Frozen human peripheral blood mononuclear cells (PBMCs) were obtained from a commercial biobank, and written informed consent was obtained by the biobank from all donors for research use of their samples. Peripheral blood samples collected from healthy volunteers were obtained following institutional guidelines, and written informed consent was provided by each volunteer prior to participation. Donor information is as follows: donor 1: male, age 60-65; donor 2: female, age 45-50; donor 3: male, 30-35. Age, sex and/or gender was not considered in the study design because this could have led to prohibitive donor identification and medical privacy breach. Participants were compensated $50- or $100-dollars USD, depending on the size of the donation.

### Fresh human PBMC cytokine production assays

Peripheral blood was collected from healthy volunteers into sodium heparinized vacutainer CPT tubes (BD Biosciences) and shaken at room temperature until centrifugation. CPT tubes were centrifuged at $1800 \times g$ for 20 min at 23 °C resulting in the separation of PBMCs from granulocytes and red blood cells. The PBMC layer was carefully transferred into a 15 mL conical tube and PBS was added to bring the total volume to 15 mL, then inverted to mix, and centrifuged at $300 \times g$ for 15 min. Supernatant was aspirated without disturbing the cell pellet, then cells were resuspended in 10 mL PBS and centrifuged at $300 \times g$ for 10 min. Supernatant was aspirated and cells were resuspended in complete medium consisting of RPMI 1640, 10% heat-inactivated FBS, and 100 U/mL Pen-Strep, transferred to T25 flasks, and rested at 37 °C with 5% $CO_2$ overnight. The following day, cells were transferred to a 15 mL conical tube, counted, and adjusted to $1E + 06$ cells/mL in complete medium. 100 uL of cell suspension ($1E + 05$ cells) was dispensed into each well of a U-bottom 96-well plate. Next, 10 mM stock compounds dissolved in DMSO were dispensed using a D300e Digital Dispenser (HP) to generate 11-point dose-response curves ranging from 1.3 nM to 50 μM, with DMSO-only representing the baseline; each point was analyzed in duplicate. All wells were normalized with DMSO equivalent to the 50 μM concentration volume and total DMSO was kept below 1% of the final volume. Plates were then incubated at 37 °C with 5% $CO_2$ for 24 h. Supernatant was collected and used for interferon-beta detection by AlphaLISA (Revvity) following manufacturer's instructions, and cell viability was assessed by CellTitre-Glo assay (Promega).

### Frozen human PBMC cytokine production assays

Frozen human peripheral blood mononuclear cells (PBMCs, BioIVT) are gently thawed, suspended in complete medium [AIMV, 10% heat-inactivated FBS, 1% antibiotic-antimycotic, 1% sodium pyruvate, 1% MEM non-essential amino acids; ThermoFisher] and concentrated by low-speed centrifugation ($400 \times g$, ambient temperature, five minutes). The cell pellet is suspended in complete medium and incubated for one h (37 °C, 5% $CO_2$) prior to measuring the cell concentration and viability. For each assay, compounds, suspended in DMSO, are first transferred to empty wells of a 384-well plate, followed by adding PBMCs suspended in complete medium to each well (100,000 cells /

well for IFN-β assays). Cells are pre-treated for one h (37 °C, 5% $CO_2$). After one h, complete medium is added to baseline control wells or 2′3′-cGAMP (100 μM, Chemietek) is added to designated 100% maximum control and test compound wells. Cells are subsequently treated for 24 h prior to transferring the supernatants to empty 384-well white, solid-bottom assay plate wells. Cellular interferon-beta was then measured using alphalisa immunoassays (Revvity) following vendor protocols. The half maximal inhibitory concentration values ($IC_{50}$; compound concentration which inhibits a response halfway between the assay baseline and maximum) are calculated using PRISM software (10.4.0).

### STING expression and purification

Human STING[155–341] G230A/R293Q, STING[155–341] H232R, STING[155–341] G230A/H232R/R293Q, STING[155–341] V155M/G230A/H232R/R293Q, STING[155–341] G158A/G230A/H232R/R293Q, and STING[155–341] G158E/G230A/H232R/R293Q were subcloned into a pET28 bacterial expression vector with a construction containing an N-terminal hexahistidine tag followed by a SUMO tag fusion and a Tobacco Etch Virus (TEV) protease cleavage site. Protein expression and purification were carried out essentially as described previously[9]. In brief, the STING expression vector was transformed into chemically competent *E. coli* One Shot BL21 Star (DE3) cells (Invitrogen). The unlabeled expression was carried out in the Terrific Broth medium (Gibco). The [13]C, [15]N labeled protein was expressed in a modified M9 medium containing D-glucose (Cambridge Isotope Laboratories, Inc., U-[13]C6, 99%) and ammonium chloride (Cambridge Isotope Laboratories, Inc., [15]N, 99%). The cells were grown in shake-flasks at 37 °C to an $OD^{600 nm}$ of 1 AU, and the expression was induced with 0.4 mM isopropyl β-D-1-thiogalactopyranoside. The induced cells were harvested after overnight incubation at 18 °C. The cells were lysed, crude protein was captured using a HisTrap FF Crude column (Cytiva), and the eluted protein was further purified using a HiLoad 26/600 Superdex 200 gel filtration column (Cytiva). Fractions from the single, major peak fractions containing enriched STING protein were pooled, and incubated with TEV protease overnight cleavage at 4 °C to cleave off the His-SUMO fusion. The untagged STING was isolated in the flow through following chromatography over Ni-NTA (Qiagen) resin. The final, purified STING protein was buffer exchanged into storage buffer (25 mM Tris 7.5, 200 mM NaCl, 0.5 mM TECP, 5% glycerol) using a PD10 column (Cytiva). Purified STING in storage buffer was flash-frozen in liquid nitrogen and stored at −80 °C until needed.

### X-ray crystallographic data collection, processing, and structure determination

Purified hSTING[155-341] G230A/R293Q was co-crystallized with diABZI-a1, diABZI-i, and THIQi as follows. STING (at a protein concentration of 10 mg/mL in 20 mM HEPES pH 7.5, 300 mM NaCl, 1 mM TCEP) was combined with ligand (final ligand concentration = 1.5 mM), and the mixture incubated on ice for 1 h. Crystallization screening was carried out by combining 0.2 μL STING + ligand mixture with an equal volume of precipitant (0.2 M either ammonium acetate or magnesium chloride hexahydrate, 0.1 M bis-tris pH 5.5-5.9, 25% (w/v) PEG 3,350) over a reservoir of 75 μL precipitant in an MRC UVXPO sitting drop vapour diffusion crystallization tray (Swissci) at either 19 °C or 4 °C. Purified hSTING[155-341] G230A/R293Q was co-crystallized with cGAMP as follows. STING (at a protein concentration of 12 mg/mL in 20 mM HEPES pH 7.5, 300 mM NaCl, 1 mM TCEP) was combined with cGAMP (final ligand concentration = 2 mM) and magnesium chloride (final $MgCl_2$ concentration = 4 mM), and the mixture incubated on ice for 2 h. Crystallization screening was carried out by combining 1 μL STING + cGAMP mixture with an equal volume of precipitant (1.1 M sodium/potassium tartrate, 0.1 M MES pH 6.0) over a reservoir of 500 μL precipitant in a 15-well hanging drop vapour diffusion crystallization tray (Nextal) at 19 °C. Purified hSTING[155-341] H232R was co-crystallized with ABZI as

follows. STING (at a protein concentration of 12 mg/mL in 20 mM HEPES pH 7.5, 300 mM NaCl, 1 mM TCEP) was combined with ABZI (final ligand concentration = 1.5 mM), and the mixture incubated on ice for 1 h. Crystallization screening was carried out by combining 0.2 μL STING + ligand mixture with an equal volume of precipitant (0.1 M tri-sodium citrate pH 5.6, 20% (w/v) PEG 4000, 20% (v/v) isopropanol) over a reservoir of 75 μL precipitant in an MRC UVXPO sitting drop vapour diffusion crystallization tray (Swissci) at 19 °C. Crystals typically formed within 24 h and grew to maximum size within 1 week. Crystals were harvested using a nylon loop, cryoprotected in precipitant supplemented with either 10% (v/v) each glycerol and ethylene glycol or 25% (v/v) glycerol and were then flash frozen in liquid nitrogen for X-ray diffraction screening. X-ray diffraction data for STING + diABZI-a1 was collected at a wavelength of 0.92 Å. and a temperature of 100 K at the National Synchrotron Light Source II (AMX beamline 17-ID-1). X-ray diffraction data for STING + ABZI, cGAMP, diABZI-i, and THIQi was collected at a wavelength of 1 Å and a temperature of 100 K at the Advanced Photon Source (IMCA-CAT beamline 17-ID). Data reduction was performed using autoPROC (Global Phasing Ltd.). Structures were phased by molecular replacement using Phaser[39] and the protein coordinates from a BMS-internal crystal structure of STING determined previously. The unpublished structure that we used for MR was itself solved by MR using the protein coordinates from RCSB PDB ID 4EMT, which was originally reported in Shu et al.[9]. Structures were completed through iterative cycles of model building using Coot[40] and restrained refinement using autoBUSTER (Global Phasing Ltd.). A summary of the data collection and refinement statistics is provided in Supplementary Table 1. The crystal structures of STING with cGAMP, diABZI-a1, THIQi, diABZI-i, and ABZI have been deposited into the RCSB PDB under accession numbers **9CUA, 9CUB, 9CUC, 9CUD**, and **9CUE**, respectively.

## NMR sample preparation
The 0.8 mM STING[155-341] G230A/H232R/R293Q -THIQi complex for backbone and methyl assignments was generated by mixing $^{13}C$, $^{15}N$-labeled STING[155-341] G230A/H232R/R293Q and THIQi (1:6) at ~10 μM concentration in 20 mM Tris-$d_{11}$ (pH 7.3) including 100 mM NaCl, 3 mM dithiothreitol-$d_{10}$, 8% $D_2O$, followed by ultrafiltration to remove extra compound. The $^{13}C$, $^{15}N$-labeled STING[155-341] G230A/H232R/R293Q -ABZI complex was made in the same way as STING[155-341] G230A/H232R/R293Q -THIQi complex, and the final concentration is ~0.95 mM. $^{2}H$, $^{13}C,^{15}N$ labeled STING[155-341] G230A/H232R/R293Q was unfolded in 20 mM Tris-$d_{11}$ (pH 7.3) including 100 mM NaCl, 8 M Urea, 3 mM dithiothreitol and followed by refolding in the presence of ABZI in the same buffer except without urea by dilution at 4 °C. The $^{2}H$, $^{13}C,^{15}N$ labeled STING[155-341] G230A/H232R/R293Q -ABZI was used for backbone assignment. Samples of the complexes were buffer exchanged to 99.96% $D_2O$ for experiments in $D_2O$.

## NMR spectroscopy and assignments
NMR experiments for assignment of THIQi and ABZI-complexed STING[155-341] G230A/H232R/R293Q were acquired at 308 K and 303 K respectively on Bruker Avance NEO-700 & 600 MHz spectrometers which both were equipped with 5 mm TCI cryo-probes. NMR data processing and analysis were performed using NMRPipe 10.9[41] and NMRFAM-Sparky 1.47[42]. Backbone $^{1}H$, $^{13}C$, and $^{15}N$ and sidechain methyl assignments of STING[155-341] G230A/H232R/R293Q -THIQi complex were obtained by analyzing 2D $^{1}H$-$^{15}N$ TROSY and $^{1}H$-$^{13}C$ constant-time HSQC (CT-HSQC) and three-dimensional HNCA, HN(CO)CA, HNCO, HN(CA)CO, $^{15}N$-edited NOESY-HSQC ($\tau_m = 60$ ms and $D_1 = 1$ s) and $^{13}C$-edited NOESY-HSQC ($\tau_m = 50$ ms and $D_1 = 1$ s) spectra collected on $^{15}N,^{13}C$ labeled sample. Backbone and methyl resonance assignments of STING[155-341] G230A/H232R/R293Q -ABZI complex were achieved by analyzing 2D $^{1}H$-$^{15}N$ TROSY and 3D HNCACB, HNCA, HN(CO)CA, HNCO, and HN(CA)CO spectra collected on $^{2}H$, $^{13}C,^{15}N$ labeled sample plus 2D

$^{1}H$-$^{15}N$ TROSY, $^{1}H$-$^{13}C$ CT-HSQC, 3D $^{15}N$-edited NOESY-HSQC ($\tau_m = 60$ ms and $D_1 = 1.1$ s), and $^{13}C$-edited NOESY-HSQC ($\tau_m = 50$ ms and $D_1 = 1$ s) spectra acquired on $^{13}C,^{15}N$ labeled sample.

In $^{1}H$-$^{13}C$ CT-HSQC experiment, methyls with an odd number aliphatic carbon neighbors show opposite intensity compared to those with an even number aliphatic carbon neighbors[43,44]. Since the $CH_3$ group of Met has no aliphatic carbon neighbors, its NMR signal intensity is opposite to those of Ala, Ile, Leu, Thr, and Val in methyl fingerprint region of NMR spectrum. Therefore, chemical shifts of Met methyls are easy to be identified. Assisted with 3D $^{13}C$-edited NOESY-HSQC, M271$^{CH3}$ and A277$^{CH3}$ in both complexes were unambiguously assigned (Fig. 2a, and Supplementary Fig. 3). The backbone and methyl assignments of STING[155-341] G230A/H232R/R293Q in complex with other ligands were achieved by comparing with 2D $^{1}H$-$^{15}N$ TROSY and $^{1}H$-$^{13}C$ HSQC of these two complexes. BMRB codes for NMR chemical shift assignments of THIQi-complexed and ABZI-complexed STING[155-341] G230A/H232R/R293Q are 52522 and 52528 respectively.

## NMR titrations
50-100 μM of $^{13}C$, $^{15}N$-labeled STING[155-341] G230A/H232R/R293Q, STING[155-341] V155M/ G230A/H232R/R293Q, STING[155-341] G158A/ G230A/H232R/R293Q, and STING[155-341] G158E/G230A/H232R/R293Q was titrated with different ligands in 20 mM Tris-$d_{11}$ (pH 7.3) including 100 mM NaCl, 3 mM dithiothreitol-d10, 8% $D_2O$, and the 2D $^{1}H$-$^{15}N$ TROSY and $^{1}H$-$^{13}C$ HSQC experiments were collected at 298 K.

## HEK cell culture and luciferase reporter assay
HEK-Lucia™ Null Cells were obtained from Invivogen (cat# hkl-null). HEK-Lucia™ cells were maintained in Dulbecco's modified Eagle's medium (DMEM) supplemented with 10% heat-inactivated fetal bovine serum (Thermo Fisher Scientific), penicillin and Normocin™ (100 U/mL and 100 ug/mL). HEK-Lucia™ cells were transfected with STING wild type, or mutant constructs using Lipofectamine™ 3000 (Thermo Fisher Scientific). After a 20 h incubation, luciferase activity was determined using QUANTI-Luc™ following the standard protocol (Invivogen).

## THP-1 cell culture and luciferase reporter assay
THP1-Dual™ KI STING cells harboring either the WT allele or V155M point mutation were acquired from Invivogen (cat# thpd-r232, thpd-m155). Both cell lines were maintained and passaged according to manufacturer's recommendation, briefly, cells were grown in RPMI 1640 supplemented with 10% heat-inactivated FBS, and Pen-Strep (100 μg/mL). Following recovery from cryopreservation ( >2 passages), cells were sub-cultured and passaged every 3 days. Selection pressure was maintained by addition of 10 μg/mL blasticidin and 100 μg/mL zeocin. All experiments were conducted on cells under 15 passages to maintain cell line stability.

Cells were seeded in 96 well plates at a density of 100,000 cells in 100 μL of media. When specified stimulation was conducted by over-laying 100 μM of 2'3' cGAMP (cat# tlrl-nacga23 dissolved in LAL water) on cells for 24 h following 1 h pre-incubation with STING inhibitor or agonist. Following incubation, plates were spun down at 500 x g for 5 mins, 20 μL of cell supernatant was collected and added to white opaque plates. 50 μL of freshly prepared QUANTI-Luc reagent 4 (cat# rep-qlc4lg1) was added to the cell supernatant and end point luminescence was read immediately on an Envision plate reader. Cell viability for each study was assessed by treating cells with CellTiter-Glo (cat# G9242) according to manufacturer's protocol and luminesce was quantified.

## THP1-Dual™ cell reporter assays
Engineered human THP1-Dual cells (Invivogen, cat# thpd-nfis), derived from the human THP1 monocyte cell line, contain two stably inte-grated inducible reporter genes measuring interferon (luciferase

production) and NF-κB (secreted embryonic alkaline phosphatase production, SEAP) promoter activities. The cells are maintained following the supplier protocols in complete culture medium [RPMI 1640, 10% heat-inactivated fetal bovine serum (FBS), 1% penicillin/streptomycin 10 ug/ml blasticidin, 1 mM sodium pyruvate, 100 ug/ml zeocin; ThermoFisher].

**Antagonist-mode.** Compounds suspended in DMSO or DMSO alone are first transferred to empty wells of a 384-well plate. Assay medium [RPMI 1640, 1% penicillin/streptomycin, 0.1% bovine serum albumin (Sigma)] is added to baseline control wells and cells suspended in assay medium are added to wells containing DMSO (100% maximum control) or test compound (15,000 per well). Cellular STING signaling is then induced by adding 2′3′-cGAMP (Chemietek), at final concentrations of 15 μM for interferon (luciferase) assays. After 24 h treatment (37 °C, 5% $CO_2$), cell viability is measured using CellTiter Fluor (Promega) and cellular luciferase is measured by adding freshly prepared Quanti-Luc (luciferase, Invivogen) following manufacturer's instructions. The half maximal inhibitory concentration values ($IC_{50}$; compound concentration which inhibits a response halfway between the assay baseline and maximum) are calculated using PRISM software (10.4.0).

**Agonist-mode.** Compounds or DMSO alone are transferred to empty wells of a 384-well plate. Assay medium is added to baseline control wells and THP1-dual cells suspended in assay medium are added to wells containing DMSO alone (100% maximum control) or test compound (15,000 per well). Interferon-beta (400 IU/mL; R&D Systems) is used to establish 100% maximum values. After 24 h treatment (37 °C, 5% $CO_2$), cell viability is measured using CellTiter Fluor (Promega) and cellular luciferase is measured using Quanti-Luc. The half maximal effective concentration values ($EC_{50}$; compound concentration which elicits a response halfway between the assay baseline and maximum) are calculated using PRISM software (10.4.0).

**Jess western**
HEK-293 Lucia cells (cat# hkl-null) were seeded in a 96 well plate at $3 \times 10^4$ cells per well. After 24 h, several variant mutants of the STING gene were transfected onto the cells using lipofectamine 3000 (Invitrogen cat. L3000001), following manufacturer's protocol. Cells were incubated overnight, then washed in 1X PBS and harvested in 50uL of Pierce IP lysis buffer (Thermo Fisher Scientific cat. 87787). Cell lysates were centrifuged to remove insoluble debris, then protein detection was performed using Protein Simple Jess western blot instrument, according to manufacturer's protocol. Samples were tested using antibodies pTBK-1 (CST, #5483, D52C2, 1:10 dilution) along with α-tubulin antibody (Novus Biologicals, NB100-690, DM1A, 1:50 dilution) as a loading control.

**Western blotting**
Cells were washed in 1X PBS and harvested in 50uL of RIPA buffer (Sigma, R0278). Cell lysates were centrifuged, and protein concentration was quantified using the Pierce BCA protein assay kit (Thermo Scientific, 23225). Following reconstitution with 4X sample loading buffer (Licor, 928-40004), 20 μg of protein samples were resolved using 4–20% Criterion precast gels (Biorad, 5671094). The protein was subsequently transferred onto a nitrocellulose membrane using the Trans-Blot turbo midi 0.2 μm Nitrocellulose transfer packs (Biorad, 1704159). To minimize the non-specific binding sites, the membrane was incubated in blocking buffer for 1 h. The blots were then probed overnight at 4 C with STING rabbit mAb (Cell Signaling, 13647, D2P2F, 1:1000 dilution) vinculin mAb (Santa Cruz, sc-73614, 7F9, 1:500 dilution). Following day, the blots were washed thrice in TBS containing 0.1% Tween 20 (TBST) before being incubated with an IRDye® 800CW (Licor, 926-32211) secondary antibody for 1 h at RT. After three washes with TBST, protein band detection was carried out using a Li-COR Odyssey imaging system.

**Compound dosing**
When specified, STING modulating compounds were dosed using the Tecan D300e digital dispenser. Compound stocks were formulated in DMSO at 10 mM. DMSO normalization on non-treated wells was conducted by automatic dispenser.

**Reporting summary**
Further information on research design is available in the Nature Portfolio Reporting Summary linked to this article.

## Data availability
Source data is available with this paper as a Source Data file. Crystallographic structure data (atomic coordinates and structure factors) have been deposited into the Protein Data Bank (https://www.rcsb.org) and are publicly available as of the date of publication. Accession numbers are: 9CUA (Human STING G230A/R293Q variant bound to cGAMP); 9CUB (Human STING G230A/R293Q variant bound to diABZI-a1); 9CUC (Human STING G230A/R293Q variant bound to THIQi); 9CUD (Human STING G230A/R293Q variant bound to diABZI-i); and 9CUE (Human STING H232R variant bound to ABZI). NMR chemical shift assignments of THIQi-complexed and ABZI-complexed STING G230A/H232R/R293Q variant have been deposited in the BMRB database (https://bmrb.io) under accession code 52522 and 52528 respectively. The data are available as of the date of publication. The PDB codes of the previously published structures used in this study are 4LOH, 4EMU, 4EMT, 4F5Y, 4EF5, 4EF4, 4F9E, 4F9G, 4F5W, 4F5E, 4F5D, 6DXG, 6DXL, 4KSY, 6DNK, 6CY7, 6CFF, 7T9V, 7T9U, 7SSM, 7MHC, 8STH, 8STI, 6MX3, 6MXE, 6NT5, 6NT7, 8GT6. Source data are provided with this paper.

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

## Acknowledgements

The authors thank Bill Degnen and Cynthia Hendrix for in vitro cellular assay support as well as our Cell Technologies and Compound Management Groups for routinely providing cell and compound supplies support. We thank Dr. Arvind Mathur for supporting the study. We also thank Mary Struthers for her insights into the paper as well as Sophie Roy for insightful conversations on the topic.

## Author contributions

S.C.W., T.X. and A.J.D. conceived the study. P.Z., J.T. and J.A.N. performed protein expression and purification. T.X. prepared NMR samples and acquired NMR experiments. M.R., D.C. and J.S. crystallized protein-ligand samples and determined crystal structures. T.X., J.C. and L.M. (Luciano Mueller) analyzed NMR data. J.N., C.X., C.M., and S.C.W. devised, developed, and executed all experiments with PBMCs. L.M. (Leidy Merselis), L.B.S., A.C., and D.L.H. performed all biochemical experiments and those involving immortalized cell lines. S.C.W., T.X. and D.C. drafted the manuscript. All authors reviewed and approved the manuscript.

## Competing interests

The authors declare no competing interests. Requests for materials should be addressed to Tao Xie.
