## [Transparent Peer Review file · Nature Communications]

Orthosteric STING inhibition elucidates molecular correction of SAVI STING

Corresponding Author: Dr Stephen Wilson

Version 0:

Reviewer comments:

Reviewer #1

(Remarks to the Author)

In this manuscript, the authors reported the critical conformational change of STING upon its activation and inactivation, which has not been captured by x-ray crystallography. They discovered that M271CH3 NMR chemical shift is a unique signature associated with the conformational change.

Given that a variety of autoinflammatory and neurodegenerative diseases are associated with dysregulated STING activation, the development of STING agonist is an important and urgent issue. Thus, the study is expected to have a broad impact not only on cellular biochemistry, but on pharmaceutical and medical sciences.

The authors might find the following specific comments useful to improve the study.

1. Several major variants, such as HAQ, H232 are known in human STING. They are indicated to have distinct affinities to cGAMP. In Figure 1a, the authors should investigate and describe the genotype of STING in healthy volunteers.
 2. Figure 3d: the precise description for 0 ng, 5 ng, 50 ng is necessary (the amount of plasmid, I guess). The enhancement of IFN activity in M271I compared to WT appears very subtle. The authors should describe the data description carefully (line 165). Need statistic significance.
- The authors should perform biochemical analysis (western blot for pTBK1, pSTING, pIRF3) and cell biological analysis (subcellular localization change of STING from the ER to the Golgi) for all of the M271 mutants shown in Figure 3d, which is a routine to investigate STING activation. The reporter assay alone is not convincing. For immunofluorescence analysis, because the trans-Golgi network (TGN) is the site of TBK1 activation (PMID: 38212328, 35125375), TGN46 (a TGN protein) can be ideally used as a co-localization marker.
3. Figure 4d: Need precise description of WT THP-1 cells and V155M THP-1 cells. THP-1 cells harbor biallelic HAQ alleles. These cells used in Figure 4d are knocked-out in HAQ alleles and express WT (R232) or V155M STING variant?
 4. Figure 4e: Does THIQi treatment (over 10 μ M) suppress STING trafficking out of the ER? As I pointed out above, WB and immunofluorescence analysis should be provided.
 5. Texts are too small to read in several Figures, which should be corrected.

Reviewer #2

(Remarks to the Author)

Xie et al. here presented crystal structures of human STING155-341 complexed with small-molecule drugs, diAZBI-a1, diABZI-i, and THIQi. These structures demonstrated that both orthosteric inhibitor and activator did not induce the β -strand lid closure, which is the key structural feature for natural ligand-mediated STING activation. Instead, the authors utilized NMR technique to identify the important residue M271 of human STING to explain inhibition or activation driven by diABZI-derived compounds.

In my opinions, the significant NMR chemical shifts of M271 do not sufficiently explain how this residue dictate the activation or inhibition of human STING. Due to the difficulties in crystallization of full-length human STING, I recommend that the authors use alternative structure-based methods, such as cryo-EM and small-angle X-ray scattering, to provide more solid evidence regarding how M271 and its mutants affect the conformations of full-length human STING, and thus activating or inhibiting the downstream signaling.

In a similar study (<https://doi.org/10.1016/j.celrep.2014.08.010>), Gao et al. provided detailed analysis of the STING-induced production of type I IFN and proinflammatory cytokine and chemokines using a series of STING mutants to access the effect of a small-molecule drug, DMXAA. Therefore, to merit publication in Nature Communications, I recommend the authors provide more functional validations and/or animal experiments to claim the observed critical interactions as the molecular diagnostic tool or potential therapeutic strategy for treating V155M SAVI patients using orthosteric STING inhibitors.

Reviewer #3

(Remarks to the Author)

The manuscript by et al. focuses on an interesting signaling regulator involved in host immune response to infection. Inhibiting STING-mediated signaling reduces interferon production while activating STING-mediated signaling promotes interferon production, therefore providing a direct link to inflammation. Interest in STING inhibition can be used to treat autoimmune diseases and inflammation, which makes this manuscript clinically relevant.

The integration of X-ray crystallography, NMR, and cell-based assays provides a provocative understanding of STING function that could not be gained by any of these alone and appears very solid. The limitations of X-ray crystallography were well circumvented with NMR studies that identified M271 as a crucial prognostic of STING activity (either reduced or activated, depending on its methyl chemical shift). It seems this is an ALLOSTERICALLY coupled site, although not extraordinarily far from the active site that may be the reason the authors have not explicitly discussed this site as allosteric. Nonetheless, its proximity to the V155 site, a disease inducing mutation, clearly illustrates that this site is both a prognostic for activity and causal in terms of activity.

While it is plausible that this site could be used to target for inhibitors, it is likely more important in the immediate future for utilizing as a diagnostic of inherent mutations. For that alone I would urge publication of this manuscript. I just have several specific questions that may help clarify the results to both NMR folks and more general readers.

1. While I believe the authors did a very nice job in convincing the ready that the M271 methyl shift is an accurate diagnostic of activity, I also wonder if there are specific shifts in the ¹⁵N-HSQC that could be used without C¹³-labeling? It may be that these are simply too small to accurately utilize, but may be worth mentioning.

2. It is also interesting to discuss a possible the relationship between the magnitude change of the M271 methyl CSP with activity. One can imagine relating this to their interferon signaling but also future studies that can probe the relationship of mutagenic shifts with downstream binding affinity (maybe TBK1?). This is NOT necessary for this paper, as the authors have really done a wonderful job thus far, but possibly just mention for future studies.

3. Figure 2e, the observation of TWO resonances of M271 can potentially be explained by binding asymmetry (I think). Meaning, I assume this is due to different chemical environments within the dimer, but I would explicitly write this for the reader to understand.

4. Figure 2g is a bit confusing, considering that you've colored M271 differently than the other resonances (I think). Maybe I would just point to M271. I'm also not really sure why the authors did this, as wasn't it already known that they both bind within the active site? Thus, the molecule with the highest affinity is expected to dominate.

5. Extended Fig. 4 is very helpful, but can we also see the CSPs versus residue in order to visualize these distally induced conformational changes to A277 and M271?

6. I'm a bit confused about the cellular assays in HEK cells, although the data largely look quite convincing. Was this utilizing a CRISPR KO of the endogenous STING? Also, in some cases activation is not additive with the hydrophobic mutations to M271 and V155, yet it some they are. I suspect this is just "normal" cell culture that can lead differential results but maybe the authors would like to comment on this?

Minor:

132-133 Should likely read: "leading to its downfield shift."

Version 1:

Reviewer comments:

Reviewer #1

(Remarks to the Author)

Reviewer1 thinks that authors tried their best to address my original critiques. Although not all of the critiques could not be addressed because of the resources that they currently have, having a new data in Supp. Figure 5 improves the quality of the manuscript.

Reviewer #2

(Remarks to the Author)

To provide a more comprehensive explanation for M271's role in dictating human STING activity, the authors have made several important modifications, including new comments in lines 278–327 of the revised manuscript, as well as the addition of Figure 6 and Supplementary Table 2. The NMR competition experiment has been conducted to further support the critical role of M271 as a unique molecular signature for STING activation, inhibition, and molecular correction. All previously raised issues have been adequately addressed, and the revised manuscript shows significant improvement over the original submission. Therefore, I support the publication of this revised manuscript in Nature Communications.

Reviewer #3

(Remarks to the Author)

In this reviewer's view, this manuscript provides novel insights into the molecular mechanisms of STING activation and inhibition, offering a diagnostic tool for STING-related diseases like SAVI, and demonstrates an example of molecular correction in a disease-associated STING mutation, potentially making it clinically relevant and of broad interest.

Response:

1.
The authors explained that using 15N-HSQC alone is challenging for a protein like STING155-341, particularly due to the crowded spectrum and the difficulty of obtaining unambiguous assignments for backbone amides. However, they did mention the identification of G278 amide as a potential probe for STING's functional state, albeit with smaller shifts than M271CH3. This addresses the reviewer's concern by confirming the difficulty of using 15N-HSQC effectively in their specific setup while providing an alternative suggestion.
 2.
The authors acknowledged the suggestion, explaining that all compounds tested bind to STING on the slow-exchange timescale, making it difficult to correlate M271CH3 shifts with interferon signaling directly. However, they proposed that it would be more feasible to explore the correlation with fast-exchange agonists and also mentioned future directions exploring the relationship between mutagenic shifts and binding affinity to downstream molecules like TBK1.
 3.
The authors clarified that the two resonances observed are indeed caused by binding asymmetry, due to different chemical environments within the dimer. They updated the manuscript to explicitly mention this in lines 135-136, improving clarity for readers.
 4.
The authors agreed with the reviewer and clarified that M271CH3 was colored differently because of intensity differences in the NMR spectrum, but this distinction caused confusion. They decided to remove Figure 2g from the manuscript, aligning with the reviewer's suggestion for a clearer presentation.
 5. Extended Figure 4 (Chemical Shift Differences vs Residue):
The authors added two figures (now Figure 3a and 3b) to illustrate both amide and methyl chemical shift differences between the STING155-341-THIQi and STING155-341-ABZI complexes. They addressed the reviewer's concern by visualizing the conformational changes more clearly.
 6. Cellular Assays in HEK Cells:
The authors clarified that they used a HEK-Luciferase reporter system with transient transfection of STING mutants to induce IFN production, rather than CRISPR KO. They also acknowledged the variability in transfection efficiency and cell culture conditions, which could explain the non-additive results. They included a STING blot to show that endogenous STING is expressed at much lower levels in HEK cells than the transiently transfected variants.
- In this reviewer's view, these revisions should effectively clarify the manuscript and improve its readability for both NMR specialists and general readers.

REVIEWER COMMENTS

We thank the reviewers for all of their constructive feedback. In response to questions and comments, we have added/rearranged the following Figures and Tables:

- Figure 3 (previously Supp. Fig. 4) in response to Reviewer #3's comments
- Added Figures 4d-g in response to Reviewer #2's comments
- Figure 6 in response to Reviewer #2's comments
- Supp. Fig. 5 in response to Reviewer #1's comments
- Supp. Table 2 in response to Reviewer #2's comments

Reviewer #1 (Remarks to the Author):

In this manuscript, the authors reported the critical conformational change of STING upon its activation and inactivation, which has not been captured by x-ray crystallography. They discovered that M271CH3 NMR chemical shift is a unique signature associated with the conformational change.

Given that a variety of autoinflammatory and neurodegenerative diseases are associated with dysregulated STING activation, the development of STING agonist is an important and urgent issue. Thus, the study is expected to have a broad impact not only on cellular biochemistry, but on pharmaceutical and medical sciences.

The authors might find the following specific comments useful to improve the study.

1. Several major variants, such as HAQ, H232 are known in human STING. They are indicated to have distinct affinities to cGAMP. In Figure 1a, the authors should investigate and describe the genotype of STING in healthy volunteers.

Our response: This is an excellent comment given the debate between natural STING variants and functional responsiveness (well-described by Sci. Rep. 2023, 13, 19541). It is important to point out that most studies that determine the affinity of cGAMP to STING variants use only the cyclic dinucleotide binding domain of the protein with the transmembrane domain cut off. In many of those studies, HAQ is a misnomer because H71 resides on the transmembrane domain, which is not present. In reality, it only reflects the AQ variant. The only study that we are aware of that uses full-length STING to compare between WT and HAQ uses a competitive radiobinding assay and reports that HAQ and WT have similar IC50s (J. Med. Chem. 2022, 65, 5675-5689, Table 2).

The study in Fig. 1a used fresh PBMCs, and unfortunately those donors are no longer with the company to enable donor recall. However, we did investigate STING genotype variation in healthy volunteers. Eighty-seven volunteers were genetically sequenced for STING with the following representation: heterozygous (56%, 49 individuals), homozygous WT (39%, 34), homozygous HAQ (3%, 3), and homozygous AQ (1%, 1).

2. Figure 3d: the precise description for 0 ng, 5 ng, 50 ng is necessary (the amount of plasmid, I

guess). The enhancement of IFN activity in M271I compared to WT appears very subtle. The authors should describe the data description carefully (line 165). Need statistic significance.

Our response: We clarified in the Figure 3d (now Figure 4h) legend that the amounts indicated refer to the amount of plasmid transiently transfected.

We also performed statistical significance on the data and updated Figure 3d. Upon review of statistical analysis, M271I does not demonstrate a statistically significant difference between WT (as suspected by the reviewer). In accord, we adjusted the data analysis and adopted more careful language on line 222-224 to reflect these changes.

The authors should perform biochemical analysis (western blot for pTBK1, pSTING, pIRF3) and cell biological analysis (subcellular localization change of STING from the ER to the Golgi) for all of the M271 mutants shown in Figure 3d, which is a routine to investigate STING activation. The reporter assay alone is not convincing. For immunofluorescence analysis, because the trans-Golgi network (TGN) is the site of TBK1 activation (PMID: 38212328, 35125375), TGN46 (a TGN protein) can be ideally used as a co-localization marker.

Our response: We thank the reviewer for recommending experiments that will make this paper stronger, but we would like to note that others have reported the effects of STING mutagenesis with only using an IFN reporter (PMID: 21947006, 33833757). We performed pTBK1 western blot analysis in HEK293 cells transiently transfected with the M271 mutants. However since the cells are transiently transfected and are also given no cGAMP agonist, we could not detect pTBK1 in any of the samples by traditional Western blot even with our V155M positive control (see below). Note: this antibody has been reliably used to detect pTBK1 upon cGAMP stimulation, which rules out issues with the antibody.

Instead, we ran the same samples on a Jess western instrument, which can detect protein at a much lower level than a traditional Western blot. We added this blot (see below) in Supp. Fig. 5, and the results largely align with our findings using the IFN reporter.

As such, we tried the same approach to look at pSTING and pIRF3, but unfortunately these were both below the limit of detection for the Jess. The logical path forward would be to create stable cellular lines that could enhance the signal for Western analysis and also be used for microscopy studies. However, we do not have the resources that are required to create stable lines. Many authors listed in this manuscript have recently left the company or been laid off, prohibiting any further biochemical/cellular experiments.

3. Figure 4d: Need precise description of WT THP-1 cells and V155M THP-1 cells. THP-1 cells harbor biallelic HAQ alleles. These cells used in Figure 4d are knocked-out in HAQ alleles and express WT (R232) or V155M STING variant?

Our response: This information can be found under “THP-1 cell culture and luciferase reporter assay” in the Methods portion of the paper. We used THP1-Dual™ KI STING cell lines (Invivogen, cat# thpd-m155) that harbor either the WT allele or V155M point mutation and have the endogenous STING alleles knocked out. We have updated Figure 4d (now Fig. 5d) to also include this information in the legend.

4. Figure 4e: Does THIQi treatment (over 10 uM) suppress STING trafficking out of the ER? As I pointed out above, WB and immunofluorescence analysis should be provided.

Our response: As previously described in #2, we no longer have the resources to enable these experiments.

5. Texts are too small to read in several Figures, which should be corrected.

Our response: We have enlarged the Figure text in many Figures to become easier to read.

Reviewer #2 (Remarks to the Author):

Xie et al. here presented crystal structures of human STING¹⁵⁵⁻³⁴¹ complexed with small-molecule drugs, diAZBI-a1, diABZI-i, and THIQi. These structures demonstrated that both orthosteric inhibitor and activator did not induce the β -strand lid closure, which is the key structural feature for natural ligand-mediated STING activation. Instead, the authors utilized NMR technique to identify the important residue M271 of human STING to explain inhibition or activation driven by diABZI-derived compounds.

In my opinions, the significant NMR chemical shifts of M271 do not sufficiently explain how this residue dictate the activation or inhibition of human STING. Due to the difficulties in crystallization of full-length human STING, I recommend that the authors use alternative structure-based methods, such as cryo-EM and small-angle X-ray scattering, to provide more solid evidence regarding how M271 and its mutants affect the conformations of full-length human STING, and thus activating or inhibiting the downstream signaling.

In a similar study (<https://doi.org/10.1016/j.celrep.2014.08.010>), Gao et al. provided detailed analysis of the STING-induced production of type I IFN and proinflammatory cytokine and chemokines using a series of STING mutants to access the effect of a small-molecule drug, DMXAA. Therefore, to merit publication in Nature Communications, I recommend the authors provide more functional validations and/or animal experiments to claim the observed critical interactions as the molecular diagnostic tool or potential therapeutic strategy for treating V155M SAVI patients using orthosteric STING inhibitors.

Our response: We appreciate the concern around providing a more comprehensive explanation for M271's role in dictating STING activity and have addressed this as follows. In the present study, we used NMR spectroscopy to reveal good correlation between M271^{CH₃} chemical shift and STING function, which was not able to be resolved by either x-ray crystallography or cryo-EM studies. This study also explained how gain-of-function mutations V155M and G158A activate STING in a ligand-independent way by shifting the conformation from the inactive state towards the active state. Furthermore, our results revealed structural properties critical for STING modulation. Although STING LBD was used in NMR study, the M271^{CH₃} chemical shifts correlate very well with both ligand binding-induced and genetically driven function states, which were characterized using full-length STING in the cell, suggesting the dimer interface conformations described in Fig. 3e-g can be predicted from the chemical shift of M271^{CH₃} and relate to the activation and inhibition of human STING.

Regarding the request for characterization using orthogonal structure-based approaches: differentiating the modest local conformational change described in Fig. 3e-g using cryo-EM will be quite challenging, given the small size of even full-length STING¹⁻³⁴¹, arduousness as a membrane protein, and expected modest resolution in cryo-EM of only 3.3-4 Å (cite Nature (2019) & (2022), Cell Disc. (2022) and Mol. Cell (2023) papers). Indeed, small-angle X-ray scattering, too, will afford too low of resolution (~10 Å maximum) to be able to differentiate the two conformations of M271 described in Fig. 3e-g. However, we believe the results presented in the present study, together with those from previously reported cryo-EM studies, afford the following explanation. The cryo-EM structure of chSTING without the c-terminal tail (CCT) reveals STING stays in an auto-inhibited state by forming a bilayer assembly packed in a head-

to-head and side-by-side way⁴⁵. The side-by-side packing is mainly mediated by LBD α 2-loop-LBD α 3 and is crucial for keeping STING in an autoinhibited state. The hydrophobic interactions in the dimer interface were suggested to play an important role in maintaining the configuration of LBD α 2-loop-LBD α 3⁴⁵. Therefore, the conformational change as identified in our NMR study in this hydrophobic region induced either by agonist binding or mutation could regulate STING function by perturbing the structure of LBD α 2-loop-LBD α 3. (lines 318-327).

In addition, we performed an NMR competition experiment to elucidate the SAVI mutation V155M functions by perturbing the conformational equilibrium between the two states (Fig. 4d-g) identified by M271^{CH3} chemical shifts. We also discuss the dimer interface conformation observed in other published PDB structures of STING in our revised manuscript (lines 278-318, Fig 6. and Supp. Table 2). Our analysis establishes a correlation between the dimer interface conformation and the open/closed conformation, suggesting the conformation of the hydrophobic region can be predicted from M271^{CH3} chemical shifts and is critical for STING regulation.

Reviewer #3 (Remarks to the Author):

The manuscript by et al. focuses on an interesting signaling regulator involved in host immune response to infection. Inhibiting STING-mediated signaling reduces interferon production while activating STING-mediated signaling promotes interferon production, therefore providing a direct link to inflammation. Interest in STING inhibition can be used to treat autoimmune diseases and inflammation, which makes this manuscript clinically relevant.

The integration of X-ray crystallography, NMR, and cell-based assays provides a provocative understanding of STING function that could not be gained by any of these alone and appears very solid. The limitations of X-ray crystallography were well circumvented with NMR studies that identified M271 as a crucial prognostic of STING activity (either reduced or activated, depending on its methyl chemical shift). It seems this is an ALLOSTERICALLY coupled site, although not extraordinarily far from the active site that may be the reason the authors have not explicitly discussed this site as allosteric. Nonetheless, its proximity to the V155 site, a disease inducing mutation, clearly illustrates that this site is both a prognostic for activity and causal in terms of activity.

Our response: We agree with the reviewer that the conformational change induced by agonist binding is allosteric. We added multiple lines within the main text to highlight that gain-of-function mutations affect conformational ensemble of STING by directly perturbing the hydrophobic interaction of this region, and that agonist binding does it in an allosteric way (line numbers 170-172, 141-143).

While it is plausible that this site could be used to target for inhibitors, it is likely more important in the immediate future for utilizing as a diagnostic of inherent mutations. For that alone I would urge publication of this manuscript. I just have several specific questions that may help clarify the results to both NMR folks and more general readers.

1. While I believe the authors did a very nice job in convincing the ready that the M271 methyl shift is an accurate diagnostic of activity, I also wonder if there are specific shifts in the ^{15}N -HSQC that could be used without ^{13}C -labeling? It may be that these are simply too small to accurately utilize, but may be worth mentioning.

Our response: Thank you for your positive feedback and thoughtful question. Three criteria must be satisfied for an effective NMR probe describing different conformational states: (1) the resonances should be in a less crowded spectrum area; (2) they should be far away from the ligand binding site; and (3) there should be reasonably large chemical shift difference between different function states. As mentioned in the main text and methods section, we only assigned backbone and sidechain methyls for STING-THIQi (inhibitor) and STING-ABZI (activator) complexes. For a protein like STING¹⁵⁵⁻³⁴¹ dimer with a molecular weight of 44 kDa, its ^{13}C -HSQC spectrum is much better than ^{15}N -HSQC spectrum. Therefore, it is more challenging for us to compare the backbone amide chemical shifts for each residue unless they are in a well-separated spectrum area and their chemical shifts can be unambiguously assigned by comparison to those of STING-THIQi and STING-ABZI complexes. However, we did identify the chemical

shift of G278 amide as another probe for predicting the functional state of STING (see Figure below). Although the chemical shift difference between active and inactive states is much smaller than that of M271^{CH3} and A277^{CH3}, G278 is close to M271 and A277 and far away from the ligand binding pocket. We did not mention G278 in the manuscript because M271^{CH3} and A277^{CH3} show much larger chemical shift changes, and these changes can be structurally explained very well for M271^{CH3} and A277^{CH3}.

2. It is also interesting to discuss a possible the relationship between the magnitude change of the M271 methyl CSP with activity. One can imagine relating this to their interferon signaling but also future studies that can probe the relationship of mutagenic shifts with downstream binding affinity (maybe TBK1?). This is NOT necessary for this paper, as the authors have really done a wonderful job thus far, but possibly just mention for future studies.

Our response: Thanks for your insightful comments. We appreciate the suggestion to explore the correlation between the magnitude of M271^{CH3} chemical shift differences and interferon signaling as well as potential future studies on the relationship between the allosteric shifts caused by mutation and binding affinity to downstream molecule like TBK1. Since all compounds tested in this paper bind to STING on the slow-exchange timescale, it is challenging to establish a relationship between the magnitude of M271^{CH3} chemical shift change and interferon signaling. However, it would be more feasible to compare the M271^{CH3} shifts of agonists that bind to STING on the fast-exchange timescale with their interferon signaling. Similarly, correlating the mutagenic shifts to interferon signaling is only possible when the conformational exchange between the active and inactive states of the mutant occurs on the fast-exchange timescale. Exploring the correlation between mutagenesis-induced shifts and binding affinity to the downstream protein TBK1 represent an exciting avenue for our future research. We will consider all these aspects in our futures studies to further elucidate the functional implication of these chemical shift changes. We have mentioned this as a potential direction for future study in the discussion section of the revised manuscript (lines 325-327).

3. Figure 2e, the observation of TWO resonances of M271 can potentially be explained by binding asymmetry (I think). Meaning, I assume this is due to different chemical environments within the dimer, but I would explicitly write this for the reader to understand.

Our response: The observation of two resonances of M271^{CH3} in the BMS-025- and cGAMP-complexed STING is indeed caused by the binding asymmetry, due to slightly different chemical environments in the dimer. We have clarified this point in the main text (lines 135-136) of the revised manuscript to enhance the clarity for the readers.

4. Figure 2g is a bit confusing, considering that you've colored M271 differently than the other resonances (I think). Maybe I would just point to M271. I'm also not really sure why the authors did this, as wasn't it already known that they both bind within the active site? Thus, the molecule with the highest affinity is expected to dominate.

Our response: We agree with the reviewer that the coloring of M271^{CH3} is confusing and Figure 2g does not provide additional information. The reason for coloring M271^{CH3} differently is that in the ¹H-¹³C CT-HSQC experiment, the methyls with an odd number of aliphatic carbon neighbors show opposite intensity compared to those with an even number aliphatic carbon neighbors. Since the CH3 group of Met has no aliphatic carbon neighbors, its NMR signal intensity is opposite to those of Ala, Ile, Leu, Thr, and Val in methyl fingerprint region of NMR spectrum. However, differentiating these resonances by color leads to confusion and is not necessary. Therefore, we agree that coloring all resonances the same and labeling M271^{CH3} is a better way for illustration. After further consideration, we agree with the reviewer that Fig 2g does not provide extra information, so we have removed it from the main text in the revised manuscript.

5. Extended Fig. 4 is very helpful, but can we also see the CSPs versus residue in order to visualize these distally induced conformational changes to A277 and M271?

Our response: We have included two figures to illustrate amide (Fig. 3a) and methyl (Fig. 3b) chemical shift differences between STING¹⁵⁵⁻³⁴¹-THIQi and STING¹⁵⁵⁻³⁴¹-ABZI complexes versus residue. Notably, chemical shifts of both M271^{CH3} and A277^{CH3} (Supp. Fig 4b) exhibit significant difference between STING¹⁵⁵⁻³⁴¹-THIQi and STING¹⁵⁵⁻³⁴¹-ABZI complexes.

6. I'm a bit confused about the cellular assays in HEK cells, although the data largely look quite convincing. Was this utilizing a CRISPR KO of the endogenous STING? Also, in some cases activation is not additive with the hydrophobic mutations to M271 and V155, yet in some they are. I suspect this is just "normal" cell culture that can lead differential results but maybe the authors would like to comment on this?

Our response: The HEK-luciferase reporting system used in Figure 3d (now Figure 4h) relies on transient transfection of STING mutants that are constitutively active to induce IFN production, which are readout by the luciferase IFN-driven reporter. In this case, endogenous STING is expressed in HEK293s but it is significantly less than what is expressed during transient transfection (see STING blot now added to Supp. Fig. 5). However, since STING signaling is

driven by STING oligomerization, constitutively active mutations can precipitate activation by oligomerization in lieu of cGAMP activation.

The reviewer is correct in his/her hypothesis that the lack of additivity is likely due to variation in cell culture. The data shown here is reflective of 3 biological replicates performed on 3 different days. Transfection efficiency is variable from experiment to experiment (reflective of experimental conditions and passage number of cells), from plate well to plate well, and there is a ceiling in the total percentage of cells transfected that depends on the cell line. All these factors can contribute to differential results.

Minor:

132-133 Should likely read: “leading to its downfield shift.”

Our response: We have changed to “leading to its downfield shift.” in the revised manuscript.